# ToolOrchestra: Elevating Intelligence via Efficient Model and Tool Orchestration

Hongjin Su [* 1 2]   Shizhe Diao [* 1]   Ximing Lu [1]   Mingjie Liu [1]   Jiacheng Xu [1]   Xin Dong [1]   Yonggan Fu [1]
Peter Belcak [1]   Hanrong Ye [1]   Hongxu Yin [1]   Yi Dong [1]   Evelina Bakhturina [1]   Tao Yu [2]   Yejin Choi [1]   Jan Kautz [1]
Pavlo Molchanov [1]

## Abstract

Large language models are powerful generalists, yet solving deep and complex problems such as those of the Humanity's Last Exam (HLE) remains both conceptually challenging and computationally expensive. We show that small orchestrators managing other models and a variety of tools are able to both push the upper bound of intelligence and improve efficiency in solving difficult agentic tasks. We introduce ToolOrchestra, a method for training small orchestrators that coordinate the use of intelligent tools. ToolOrchestra makes explicit use of reinforcement learning with outcome-, efficiency-, and user-preference-aware rewards. Using ToolOrchestra, we produce Orchestrator, an 8B model that achieves higher accuracy at lower cost than previous tool-use agents while aligning with user preferences on which tools are to be used for a given query. On HLE, Orchestrator achieves a score of 37.1%, outperforming GPT-5 (35.1%) while being 2.5x more efficient. On $\tau^2$-Bench and FRAMES, Orchestrator surpasses GPT-5 by a wide margin while using only about 30% of the cost. Extensive analysis shows that Orchestrator achieves the best trade-off between performance and cost under multiple metrics, and generalizes robustly to previously unseen tools. These results demonstrate that composing diverse tools with a lightweight orchestration model is both more efficient and more effective than existing methods, paving the way for practical and scalable tool-augmented reasoning systems.

## 1. Introduction

Large language models (LLMs) have been reported to have made remarkable strides towards superhuman intelligence but remain of limited utility in complex agentic tasks such as those posed by the Humanity's Last Exam (HLE) (Phan et al., 2025). Tool use is a promising avenue for the extension of their capabilities beyond what can be learned from the training data. By calling on external resources through search engines and code interpreters, tool use has been shown to enhance accuracy and reduce hallucinations (Qin et al., 2023; Schick et al., 2023; Qin et al., 2024; Gehring et al., 2024; Qian et al., 2024; Yu et al., 2024; Goldie et al., 2025; Zhang et al., 2025a; Qian et al., 2025a).

Prior research on tool-use agents has primarily focused on equipping a single powerful model with utility tools such as web search or calculators. While effective in many scenarios, this approach underutilizes the potential of tools: humans, when reasoning, routinely extend themselves by calling upon resources of greater-than-human intelligence, from domain experts to sophisticated processes and software systems. Motivated by this observation, we propose the *orchestration paradigm*. Under this paradigm, intelligence emerges not from a monolith but from a composite system. At the center of the system lies an *orchestrator* model, whose responsibility is to invoke the right tools for the given task, and to do so in the right order to accomplish the task. The crucial difference to the standard monolithic setup featuring a single powerful model is that in addition to deterministic utilities such as web search functions and code interpreters, models of various capabilities are made available to the orchestrator as *intelligent tools*. The use of tools of different levels of intelligence comes at varying costs, and the challenge for the orchestrator is then to dynamically decide on which tools to invoke in order to solve the task while respecting user preferences for various tools and minimizing the cost. By delegating narrowed-down sub-problems of a larger effort requiring intelligence to intelligent tools instead of handling the entire effort by a single generalist, orchestration teems with the promise of exhibiting higher intelligence than any of the system's tools and leading monolithic solutions alike.

One approach to implementing the orchestrator paradigm is to employ a language model as the orchestrator and allow it to invoke stronger models only when it deems it necessary. This can be done naively by *prompting* an off-the-shelf

---

[*]Equal contribution  [1]NVIDIA  [2]The University of Hong Kong. Correspondence to: Hongjin Su <hjsu@cs.hku.hk>.

*Proceedings of the $43^{rd}$ International Conference on Machine Learning*, Seoul, South Korea. PMLR 306, 2026. Copyright 2026 by the author(s).

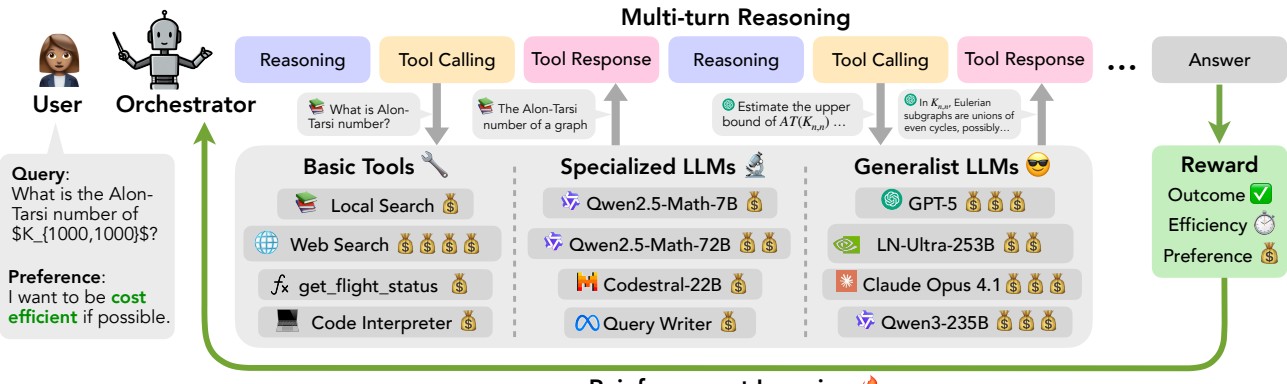

*Figure 1.* Overview of Orchestrator. Given a task, Orchestrator alternates between reasoning and tool calling in multiple turns to solve it. Orchestrator interacts with a diverse tool set, including basic tools (web search, functions such as `get_flight_status`, etc.), specialized LLMs (coding models, math models, etc.) and generalist LLMs (GPT-5, Claude Opus 4.1, etc.). In training under ToolOrchestra, Orchestrator is jointly optimized by outcome, efficiency and preference rewards via reinforcement learning.

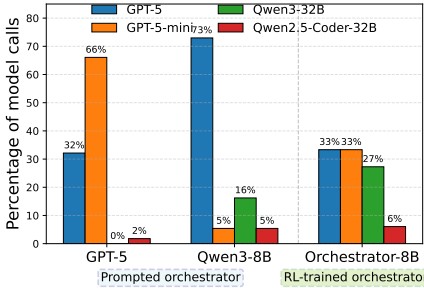

*Figure 2.* Tool-calling preferences exhibited by a prompted off-the-shelf or RL-trained model. GPT-5 tends to call GPT-5-mini most of the time, while Qwen3-8B relies heavily on GPT-5.

language model or by *training* a general-purpose orchestrator. For the former, we find that relying on straightforward model prompting is brittle and introduces systemic biases. As shown in Figure 2 (left and middle), GPT-5 disproportionately delegates tasks to GPT-5-mini, while Qwen3-8B defers to GPT-5 at a markedly higher rate. This illustrates two present issues of prompting in the context of complex tool orchestration: (i) the overuse of developmentally-related variants of oneself, i.e., *self-enhancement bias* (Zheng et al., 2023), and (ii) defaulting to the strongest available tool regardless of the cost or relative utility (see Appendix A for more details and §4 for a thorough comparison to baselines). As such, we conclude that the scenarios in which an orchestrating model may call on models and tools of capabilities both inferior and superior to its own are idiosyncratic in the context of model tool calling and warrant their own approach to training. In addition, controllability in tool-use agents remains underexplored along two axes: cost–efficiency and user preferences (cf. §7).

We address these shortcomings by proposing ToolOrchestra (shown in Figure 1), a novel method for training a small language model to act as the orchestrator – the "brain" of a heterogeneous tool-use agent. Using ToolOrchestra, we produce the Orchestrator, an 8B-parameter model trained

end-to-end with reinforcement learning (RL) to decide when and how to invoke more intelligent language models and various tools such as web search or code interpreters, and how to combine them in multi-turn reasoning. Our reward design balances three objectives – correctness of the final outcome, efficiency in resource usage, and alignment with user preferences – to yield a cost-effective and user-controllable tool-use policy. To aid RL training, we build an automatic data synthesis pipeline that generates thousands of verifiable multi-turn tool-use training examples with complex environments across 10 domains. We will make the resulting dataset, ToolScale, publicly available to facilitate further research on tool-use agent training.

In our experiments, we rigorously evaluate the merits of our approach on three challenging tasks. On HLE (Phan et al., 2025), a benchmark consisting of difficult questions across many disciplines, we find that Orchestrator substantially outperforms prior methods with far lower computational cost. We also test on $\tau^2$-Bench (Barres et al., 2025), a function-calling benchmark, where Orchestrator demonstrates the ability to schedule a variety of tools effectively, calling a large model (GPT-5) in only ∼40% of the steps and utilizing cheaper models or tools for the rest, yet still exceeding the performance of an agent that uses the large model for every step. Finally, additional evaluations on the FRAMES (Krishna et al., 2024), a factuality reasoning benchmark, provide further evidence of the versatility and robustness of our approach. We observe that even though the training and testing tasks differ markedly, the RL-trained Orchestrator adapts its tool-use policy to new challenges, indicating a high degree of general reasoning ability.

Our contributions can be summarized as follows: (1) We introduce ToolOrchestra, a method for training a small language model to serve as the orchestrator of a diverse toolkit, including classical tools and more intelligent models. This dovetails with recent developments in the field testifying

that small language models are often sufficiently powerful and far more economical in agentic systems (Belcak et al., 2025; Zhao et al., 2025). (2) We develop a novel reward training design that goes beyond accuracy. The resulting Orchestrator is trained end-to-end to balance task outcome correctness, efficiency in cost and latency, and alignment with user cost and tool preferences. (3) We demonstrate that Orchestrator trained by ToolOrchestra achieves state-of-the-art performance on challenging reasoning benchmarks, surpassing frontier models while using only a fraction of their compute and wall-clock time, and that it generalizes robustly to unseen tasks and tools.

## 2. Agentic Problem Formulation

### 2.1. Task Formulation

We investigate multi-turn tool-use agentic tasks and formalize them as a Markov Decision Process (MDP) $\mathcal{M} = (\mathcal{U}, \mathcal{S}, \mathcal{A}, \mathcal{O}, \mathcal{T}, \mathcal{Z}, r, \rho, \gamma)$ following conventions similar to prior work (Xi et al., 2024; Zhou et al., 2024; Xi et al., 2025). We are given an instruction $u \in \mathcal{U}$, user action preferences $p = (0 \leq p_a \leq 1 \text{ for } a \in \mathcal{A})$, an initial state drawn from $\rho(\cdot \mid u)$, an initial observation $o_0 \in \mathcal{O}$, and the environment state space $\mathcal{S}$. At step $k$, the Orchestrator chooses an action $a_k \in \mathcal{A}$ according to a policy $\pi_\theta(a_k \mid h_k)$ where $h_k = (u, o_0, a_0, o_1, \ldots, a_{k-1}, o_k)$ is the interaction history. The environment transitions according to $\mathcal{T}(s_{k+1} \mid s_k, a_k)$ and emits an observation $o_{k+1} \sim \mathcal{Z}(\cdot \mid s_{k+1}, a_k)$. The actions $a_i$ come at costs $c_i$ and operational latency $l_i$, and the alignment of each action with user preferences is $p_{a_i}$. After $N$ interaction steps, Orchestrator has traced the trajectory $\tau = h_N$ and the environment provides a reward $r(\tau) \in [0, 1]$ based on its correctness. Our goal is to maximize the correctness reward $r(\tau)$ and the overall user preference alignment $\sum p_{a_i}$ while minimizing the total cost $\sum c_i$ and the aggregate latency $\sum l_i$.

### 2.2. Multi-Turn Rollout

Given a user task, Orchestrator produces a solution via an iterative rollout that interleaves tool use with environment feedback to form a trajectory of turns. The rollout is initialized with a predefined system prompt and the question; the model (assistant role) then generates an initial step that ends with an EOS token. Each turn follows a *reasoning–action–observation* loop: (1) **Chain-of-thought (reasoning).** The Orchestrator analyzes the current state and plans the next action. (2) **Tool call (action).** Based on its reasoning, Orchestrator selects a tool from the available set (e.g., APIs, specialized models, code interpreters) and specifies parameters. (3) **Tool response (observation).** If a tool call is present, the tool-call block is extracted and executed by the environment; the resulting output is appended to the context under the user role and fed back to the model

for the next turn. This process repeats until Orchestrator receives a termination signal from the environment or the rollout reaches a maximum of 50 turns.

## 3. ToolOrchestra

Our approach, ToolOrchestra, centers on training a small language model as an intelligent agentic model capable of solving complex tasks by dynamically selecting and utilizing a wide variety of external tools. It can be viewed as a four-sided decision making problem, where the orchestrator needs to align with the user, tools, models and the generation by itself. We leverage RL to train models to harmonize multiple objectives when making decisions in a complex scenarios. We hypothesize that small language models suffice for this purpose if they are taught to coordinate more intelligent tools strategically, and thus choose to train an 8B model. ToolOrchestra consists of an end-to-end reinforcement learning setup where the model under training, termed Orchestrator, learns to generate optimal multi-step reasoning and tool-use trajectories. The overall architecture is illustrated in Figure 1.

### 3.1. Unified Tool Calling

In contrast to prior tool-use agents (Li et al., 2025b; Jin et al., 2025), we broaden the toolset to include domain-specialized models and expose all tools through a single, unified interface. Tools are specified in JSON as a list of objects; each object defines the tool name, description, and a typed parameter schema (names and descriptions). When LLMs are used as tools, we obtain their descriptions with the following steps: (1). randomly sample 10 training tasks; (2). obtain the trajectories of LLMs to finish these tasks; (3). Ask another LLM to write the description based on the task instructions, LLM trajectories and whether LLMs complete the tasks. In Appendix C, we show an example description of Qwen3-32B. The complete catalog of tools used in our training is provided in Appendix D.

### 3.2. End-to-End Agentic Reinforcement Learning

**Reward design.** We introduce outcome, efficiency and preference rewards into the training. For outcome reward, each rollout trajectory $\tau$ in a rollout batch T receives a binary accuracy reward $r_{\text{outcome}}(\tau) \in \{0, 1\}$ based on whether $\tau$ solves the task:

$$r_{\text{outcome}}(\tau) = \begin{cases} 1 & \text{if solved}(\tau), \\ 0 & \text{otherwise.} \end{cases} \quad (1)$$

We leverage GPT-5 as a judge to compare the answers, e.g., a name, a date, etc., providing greater flexibility in handling diverse predictions.

To encourage efficient solutions, we penalize the model under training for excessive compute or latency with the following rewards: $r_{\text{compute}}(\tau) = -\$(\tau)$, $r_{\text{latency}}(\tau) = -Clock(\tau)$, where $\$(\tau)$ is the monetary cost of $\tau$ and

$Clock(\tau)$ is the consumed wall-clock time by $\tau$. To establish a unified measurement on the compute of both open-sourced and proprietary models, we convert both the input tokens and output tokens to monetary costs following the third-party API pricing systems. See more details in Appendix E.

Preference reward is designed to encourage models to consider user preferences when choosing tools at each step. Given a set of tools $\{t_1, t_2, ..., t_n\}$ and a rollout trajectory $\tau$, we consider the vector $M^\tau = [m_{t_1}^\tau, m_{t_2}^\tau, ..., m_{t_n}^\tau, r_{\text{outcome}}(\tau), r_{\text{compute}}(\tau), r_{\text{latency}}(\tau)]$, where $m_{t_\bullet}^\tau$ is the number of times tool $t_\bullet$ is invoked in $\tau$, $M^\tau[n+1] = r_{\text{outcome}}(\tau)$.

During RL training, we first normalize each element, and project the rewards of outcome, compute, latency and tool usage on the user preference vector. Specifically, for $M^\tau[k]$ ($1 \leq k \leq n+3$) over the rollout batch T as follows: $M_{\text{normalized}}^\tau[k] = (M^\tau[k] - M_{\text{min}}^{\text{T}}[k])/(M_{\text{max}}^{\text{T}}[k] - M_{\text{min}}^{\text{T}}[k])$, where $M_{\text{min}}^{\text{T}}[k]$ and $M_{\text{max}}^{\text{T}}[k]$ are minimum and maximum value for $M^\bullet[k]$ in the batch T. If $M_{\text{max}}^{\text{T}}[k] = M_{\text{min}}^{\text{T}}[k]$, we disregard $M^\tau[k]$ by setting it to zero. We calculate the final reward for a trajectory $\tau$ as:

$$R(\tau) = \begin{cases} M_{\text{normalized}}^\tau \cdot P & \text{if } r_{\text{outcome}}(\tau) \\ 0 & \text{otherwise.} \end{cases} \quad (2)$$

where $P = [p_{t_1}, p_{t_2}, ..., p_{t_n}, p_{\text{outcome}}, p_{\text{compute}}, p_{\text{latency}}]$ ($0 \leq p_\bullet \leq 1$) is the preference vector, indicating the extent the user would like to optimize $M[\bullet]$. For example, $P[1] = p_{t_1} = 1$ indicates strong user preference to use the tool $t_1$, while $P[n+1] = p_{\text{outcome}} = 1$ and $P[n+2] = p_{\text{compute}} = 0$ implies that the user wants to exclusively optimize accuracy without considering the computational cost.

**Training procedure.** Orchestrator is fine-tuned using a policy gradient reinforcement learning algorithm, specifically Group Relative Policy Optimization (GRPO) (Shao et al., 2024). For each task in a batch, the policy $\pi_\theta$ generates a batch of trajectories T. Each trajectory $\tau \in$ T is assigned a scalar reward $R(\tau)$ (as calculated in Equation 2), and GRPO normalizes this reward within its group to compute an advantage:

$$A(\tau) = \frac{R(\tau) - \text{mean}_{\tau \in \text{T}} R(\tau)}{\text{std}_{\tau \in \text{T}} R(\tau)}. \quad (3)$$

The policy is then updated to maximize the following:

$$\mathcal{L}_{\text{GRPO}}(\theta) = \mathbb{E}\tau \sim \pi_\theta \Big[ \min \Big( \text{ratio}_\theta(\tau) A(\tau), \\ \text{clip}(\text{ratio}_\theta(\tau), 1 - \epsilon, 1 + \epsilon) A(\tau) \Big) \Big], \quad (4)$$

where $\text{ratio}_\theta(\tau) = \frac{\pi_\theta(\tau)}{\pi_{\text{old}}(\tau)}$ is the likelihood ratio between the current and previous policy.

**Training techniques.** To stabilize RL training and avoid KL loss explosion for this agent system, we perform the following during backward propagation: (1) *homogeneity filtering*, when the standard deviation of rewards in a rollout batch is smaller than $0.1$, because this indicates that most rollouts in a batch exhibit similar behaviors, and provides weak training signals; (2) *format consistency filtering*, when the example output is not aligned with the tool call format; (3) *invalid output filtering*, when the example does not produce a valid answer or output.

### 3.3. Data Synthesis

During training, we employ a large volume of diverse and realistic public datasets from GeneralThoughtArchive [1]. Additionally, we also build an automatic pipeline to synthesize more heterogeneous data across different domains and scenarios.

**ToolScale.** To enable end-to-end RL training of Orchestrator, we require agentic tool-call tasks, but verifiable data of this kind is scarce. To generate such data, we devise a two-step process: (1) simulating rich user-agent-tool environments, including creating database schemas and tool APIs; and (2) generating diverse user tasks together with their corresponding ground truth solutions based on the environment. Figure 3 provided an overview of this process. Firstly, to simulate real-world user-agent-tool environments scalably, we choose a domain $D$ and then ask an LLM to generate a database which includes schema, major subjects to focus on and database entries (as illustrated in the top-left of Figure 3). Based on the given domain $D$, LLM proposes frequently-used tools. Secondly, to increase the diversity of the task instructions, LLM first proposes diverse intents frequently seen in domain $D$, and then convert them to specific tasks based on detailed database information. Each generated task consists of task instruction $I$, golden function calls $A = a_1, a_2, ..., a_l$, and short information $o$ that must be mentioned during the process to solve the task. To enhance the difficulty of the generated tasks, we leverage an additional LLM to complicate tasks by adding more complexities such as more constraints. To ensure the quality of the synthesized data, we filter the data to remove a task if: (1) the execution of golden function calls reports an error; (2) LLMs cannot solve it in pass@8; and (3) the task can be solved without any actions. In Table 4, we list the statistics of the generated data in each domain. More examples and prompts used to synthesize data can be found in Appendix K. To evaluate whether a trajectory $\tau$ solves the given task, we define the following criteria: (1) *execution correctness*, namely whether the database content matches after executing the golden function calls $A$ and the trajectory $\tau$; (2) *process fidelity*, i.e., whether the predefined information $o$, which is required to be communicated in the process to solve the task, is mentioned in $\tau$; (3) *operation*

---

[1] https://huggingface.co/datasets/RJT1990/GeneralThoughtArchive

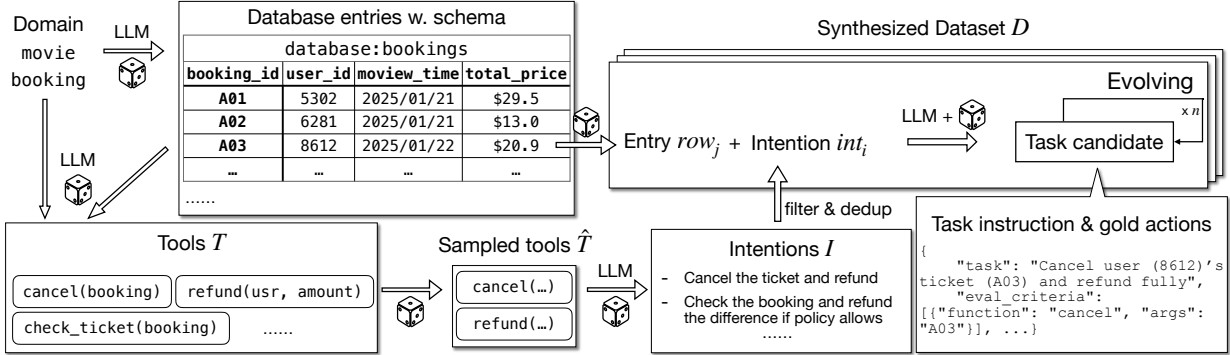

*Figure 3.* Overview of ToolScale data synthesis pipeline. Starting from a domain, LLM will (1) firstly generate domain-specific database and tool APIs to simulate the environment and (2) then generate diverse user tasks together with their corresponding golden actions.

*completeness*, that is whether the database entries operated in the ground truth trajectory $A$ are also operated in $\tau$. We consider $\tau$ to solve the task if each of these three criteria is satisfied, and fail to solve it otherwise.

**User preference.** Different users possess different preferences. For example, some users prefer local search to safeguard privacy, while others opt for internet-based search to access broader knowledge. To train Orchestrator to account for such preferences in tool selection, we construct pairs of preference instruction $PI$ and preference vectors $P$, which indicate the extent a user would like to optimize certain features, e.g., latency, or the frequency to use a particular tool (§3.3). Given a tool set $\{t_1, t_2, ..., t_n\}$, and the corresponding configuration metadata (e.g., tool price, latency), an LLM proposes diverse pairs of $(PI, P)$, which are then valiadated by another LLM to verify consistency (see Appendix F for a sample pair). The pairs are then split into two sets $Pairs_{train}$ and $Pairs_{eval}$ for training and evaluation, respectively. We concatenate the generated preference instruction to the example instruction, and augment training and testing data with user preference. During training, we use Equation 2 and the generated preference vector $P$ to calculate reward, but using Equation 6 and $P$ to calculate metrics in the evaluation. More details on rewards are included in Appendix L.

**General tool configuration.** To enhance Orchestrator's generalization abilities, we curate a diverse set of tool configurations to prevent overfitting to specific usage patterns and encourage robust, general-purpose invocation. To emulate heterogeneous user access, we randomize the subset of tools available in each training instance, encouraging Orchestrator to optimize under varying constraints rather than relying on a fixed toolkit. We also vary pricing schedules across training instances to reflect heterogeneous tool costs, exposing the model to different cost configurations so it learns to adapt its optimization strategy as prices change. In aggregate, this approach models the variability in both tool availability and cost structures across users, yielding a richer supervisory signal for optimizing Orchestrator.

# 4. Experimental Setting
## 4.1. Tools

In the training, we prepare a large and comprehensive tool set (Appendix D), but only sample a subset for each training instance to build diverse tool configurations (§3.3). We fix the following tool set in the evaluation for fair comparison.

- **Basic tools.** We use Tavily search API [2] for web search, Python sandbox for Code interpreter, and build Faiss index with Qwen3-Embedding-8B (Zhang et al., 2025b) for local search. Additionally, we also incorporate domain-specific functions, such as get_flight_status, to address specialized challenges within those domains.
- **Specialized LLMs.** We prompt GPT-5 (OpenAI, b), GPT-5-mini (OpenAI, b) as code writer, employ Qwen2.5-Coder-32B-Instruct (Hui et al., 2024) as another code writer, and leverage Qwen2.5-Math-72B (Yang et al., 2024), Qwen2.5-Math-7B (Yang et al., 2024) as specialized math models.
- **Generalist LLMs.** We consider GPT-5, GPT-5-mini, Llama-3.3-70B-Instruct (Dubey et al., 2024), and Qwen3-32B (Yang et al., 2025) as representative generalist models.

## 4.2. Baselines

We compare Orchestrator-8B produced by ToolOrchestra to baseline orchestrators constructed by prompting LLMs. Additionally, we also compare to off-the-shelf monolithic LLM systems that are (1) not equipped with tools, (2) equipped with basic tools, and (3) using the expanded tool set that further includes specialized expert models and strong generalist models.

For off-the-shelf LLMs, we evaluate GPT-5, Claude Opus 4.1 (Anthropic, 2025), Llama-3.3-70B-Instruct, Qwen3-235B-A22B (Yang et al., 2025), Llama-3_3-Nemotron-Super-49B-v1 (Bercovich et al., 2025), Qwen3-8B (Yang et al., 2025).

---

[2]https://www.tavily.com/

*Table 1.* Comparison of Orchestrator-8B with baselines (prompt-based LLMs). Llama-Nemotron-49B denotes Llama-3.3-Nemotron-Super-49B-v1. Cost in US cents, Latency in minutes, are averaged between HLE and Frames. More efficiency statistics on $\tau^2$-Bench are in Table 15 in Appendix. Basic tools include domain functions, search and code interpreter (§4.1). ↓ The lower the better.

| Tools | Model(s) | HLE (↑) | FRAMES (↑) | $\tau^2$-Bench (↑) | Cost (↓) | Latency (↓) |
|---|---|---|---|---|---|---|
| Existing baselines | GPT-5 | 35.2 | – | 84.2‡ | – | – |
| | o3 | 24.3 | – | 68.4 | – | – |
| | GPT-4o | 5.3 | – | 43.8 | – | – |
| | ToRL | 0.9 | 14.8 | 29.8 | 0.2 | 0.5 |
| | RouteLLM | 7.2 | 28.3 | 24.9 | 13.4 | 8.7 |
| No tool | Qwen3-8B | 3.2 | 24.2 | –* | 0.2 | 0.6 |
| | Llama-Nemotron-49B | 3.6 | 25.6 | –* | 0.4 | 1.1 |
| | Llama-3.3-70B | 3.8 | 32.4 | –* | 0.5 | 1.4 |
| | Qwen3-235B-A22B | 5.2 | 34.3 | –* | 2.6 | 3.3 |
| | Claude Opus 4.1 | 11.7 | 58.2 | –* | 27.4 | 8.2 |
| | GPT-5 | 23.4 | 66.3 | –* | 6.2 | 4.1 |
| Basic tools | ToolLlaMA-2-7b-v2 | 1.3 | 12.6 | 11.4 | 1.0 | 1.6 |
| | Qwen3-8B | 4.7 | 26.5 | 40.7 | 1.3 | 2.2 |
| | Llama-Nemotron-49B | 6.8 | 28.2 | 23.2 | 2.5 | 3.5 |
| | Llama-3.3-70B | 4.6 | 42.3 | 17.6 | 2.8 | 4.3 |
| | Qwen3-235B-A22B | 14.0 | 39.5 | 52.9 | 12.3 | 10.2 |
| | Claude Opus 4.1 | 19.8 | 63.5 | 46.0 | 76.2 | 32.5 |
| | GPT-5 | 35.1 | 74.0 | 77.7 | 30.2 | 19.8 |
| | RestGPT | 35.6 | 74.3 | 78.2 | 47.4 | 29.7 |
| Basic tools, Specialized LLMs Generalist LLMs | ToolLlaMA-2-7b-v2 | 18.8 | 48.6 | 52.4 | 16.2 | 11.3 |
| | Qwen3-8B | 30.6 | 68.9 | 72.3 | 27.6 | 18.3 |
| | Llama-Nemotron-49B | 25.8 | 57.9 | 66.7 | 25.6 | 17.1 |
| | Llama-3.3-70B | 19.7 | 52.4 | 55.8 | 19.7 | 13.4 |
| | Qwen3-235B-A22B | 32.8 | 74.2 | 75.6 | 29.7 | 21.2 |
| | Claude Opus 4.1 | 34.6 | 72.8 | 76.8 | 52.5 | 25.6 |
| | GPT-5 | 21.2 | 57.5 | 62.3 | 17.8 | 13.6 |
| | RestGPT | 21.6 | 57.7 | 62.8 | 27.2 | 18.9 |
| | **Orchestrator-8B** | **37.1** | **76.3** | **80.2** | **9.2** | **8.2** |

† The HLE results of Existing reported SOTA are based on the full set, while other baselines and ours are only on the text-only subset.
‡ Due to implementation differences, we could not fully reproduce GPT-5's reported result (84.2) and only reached 77.7 in our experiments.
* $\tau^2$-Bench cannot be solved in the absence of tools.

Additionally, we also compare to several existing baselines: (1). RestGPT (Song et al., 2023): a tool-use framework that prompts LLMs to conduct coarse-to-fine planning and enhances the model's ability of task decomposition and API selection; (2). ToolLlaMA-2-7b-v2 (Qin et al., 2023): an LLM trained with diverse API data to execute complex instructions; (3). ToRL (Li et al., 2025b): a model trained to discover optimal strategies for tool utilization; (4). RouteLLM (Ong et al., 2024): a router model that dynamically selects between a stronger and weaker LLM during inference.

### 4.3. Evaluation Configuration

We conduct experiments on three popular benchmarks with complex reasoning: **Humanity's Last Exam (HLE)**, **FRAMES**, and $\tau^2$-**Bench**. Details about these three benchmarks are given in Appendix B. Throughout the evaluation, we use the official price for proprietary models and leverage the pricing systems of TogetherAI[3] for open-source models. We set the inference temperature to 0 and allow maximum 50 turn for Orchestrator to solve a task. We use GPT-5 as the backbone LLM for RestGPT and leverage the default configuration for other baselines. For RestGPT and ToolLlama where the tool set can be flexibly defined, we run experiments with two settings: (1). use only basic tools; (2). used

both basic tools and specialized/generalist LLMs, which are aligned with our default setting of Orchestrator-8B.

### 4.4. Training Configuration

We employ Qwen3-8B as the backbone LLM and train it on the GeneralThoughtArchive[4] dataset in conjunction with synthetic data (§3.3). The training configuration uses a learning rate of 1e-6, a maximum input sequence length of 24,000, and a maximum generation length of 8,000, with a training batch size of 16 and a rollout batch size of 8. We allow maximum 50 turns for the Orchestrator to finish a task during rollout and use 16 NVIDIA H100 GPUs throughout the training.

## 5. Experimental Results

We compare Orchestrator against a wide range of baselines across HLE, FRAMES, and $\tau^2$-Bench. The results are summarized in Table 1. ToRL is trained to use only code interpreter and RouteLLM is trained to operate a fixed set of subordinate LLMs, which makes them less generalizable to broad tool-calling and model using scenarios. For simple prompting methods without tools, models such as Qwen3-235B-A22B and Llama-3.3-70B fail to demonstrate strong performance. This highlights the inherent difficulty

---

[3] https://www.together.ai/pricing

[4] https://huggingface.co/datasets/RJT1990/GeneralThoughtArchive

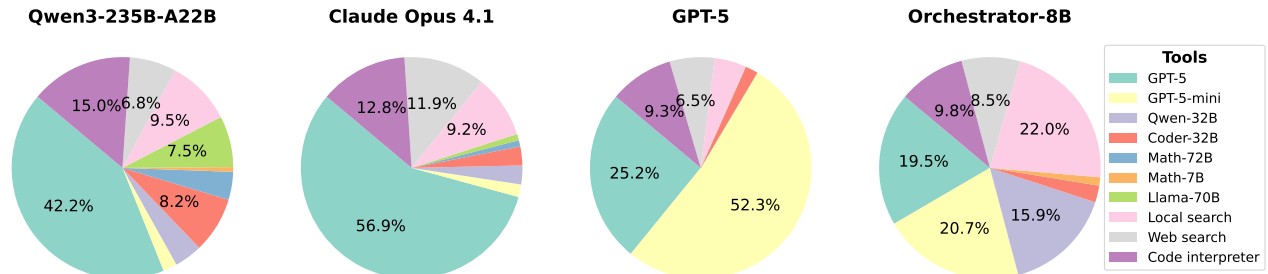

*Figure 4.* The proportion of tool calls made by LLMs to solve a task (averaged across HLE, Frames and $\tau^2$-bench). Qwen-32B refers to Qwen3-32B (Yang et al., 2025) and Coder-32B refers to Qwen2.5-Coder-32B-Instruct (Hui et al., 2024). Compared to other strong foundation models, Orchestrator-8B makes more balanced tool calls, and does not exhibit strong biases toward a particular tool or model. Detailed statistics are shown in Table 14.

of the benchmarks, where tool use or additional reasoning mechanisms is essential. Providing tool access improves performance in some cases. For instance, Claude Opus 4.1 with tool usage improves from 11.7 to 19.8 in HLE, and from 58.2 to 63.5 in FRAMES, but at the expense of 2.8x costs and 4x latency. Smaller models like Qwen3-8B perform poorly (4.7 on HLE), indicating that basic tools alone are insufficient. Combining tools with specialized and generalist LLMs generally improves results — Qwen3-235B-A22B, for example, rises from 14.0 to 32.8 on HLE and from 39.5 to 74.2 on FRAMES, but consumes more than 2 times of cost and latency. However, the gains are inconsistent across different models. GPT-5 using both tools and models suffers from performance drop due to inherent biases, often defaulting to GPT-5-mini (§6.1). RestGPT relies on prompting LLMs in complex tool calling, which may not make the best use of specialized and generalist models. ToolLlama is limited to the reasoning and long-context understanding capability, which constraints the ability to address multi-step agentic tasks that would require large context windows.

In contrast, our Orchestrator-8B achieves 37.1 on HLE and 76.3 on FRAMES, surpassing all baselines by a large margin. In $\tau^2$-Bench, Orchestrator-8B outperforms GPT-5 using basic tools by 2.5%, exhibiting strong function calling capabilities. Notably, compared to GPT-5 with tool use (35.1 on HLE) and Qwen3-235B-A22B with tool + model (32.8 on HLE), our approach delivers consistent improvements of +2 to +4.3 absolute points, while using only a small fraction of cost and time. These gains are particularly striking given that Orchestrator has only 8B parameters, but is capable of coordinating more intelligent models such as GPT-5, and achieves better performance with lower cost, which highlights the effectiveness of the orchestration strategy. Overall, the results clearly demonstrate the effectiveness of ToolOrchestra and the superiority of Orchestrator model in both performance and efficiency.

## 6. Analysis

### 6.1. Tool Use Analysis

Figure 4 shows the proportion of calls to each tool when various models serve as the orchestrator to solve a task. Instead of excessively invoking strong models and expensive tools, Orchestrator-8B learns to coordinate them more strategically. For example, in choosing between different models, Claude Opus 4.1 relies on GPT-5 most of the time, while making fewer calls to other models. In contrast, GPT-5 prefers to use GPT-5-mini. Orchestrator-8B learns to choose between various tools strategically, and achieves superior performance while using significantly lower costs.

### 6.2. Cost Analysis

To analyze the cost-effectiveness, we display the performance on HLE as a function of cost in Figure 5. We experiment with settings where the maximum number of 10, 20, 50 and 100 turns are allowed to finish a task. As the maximum number of allowed turns increases (i.e., cost increases), all models show improved performance. Orchestrator-8B consistently outperforms GPT-5, Claude Opus 4.1 and Qwen3-235B-A22B at a given budget, and can achieve similar results at a substantially lower cost. This demonstrates the capability of Orchestrator-8B to manage a heterogeneous set of tools, and pushes the intelligence boundary of the system as a whole.

### 6.3. Generalization

To evaluate Orchestrator-8B's generalization capability, we test it with a tool configuration containing models unseen during training: (1) Query writer: Claude Opus 4.1, o3-mini and GPT-4o (OpenAI, a); (2) Code writer: Claude Opus 4.1, Claude Sonnet 4.1 and Codestral-22B-v0.1 (team, 2024); (3) Math model: OpenMath-Llama-2-70b (Toshniwal et al., 2024), DeepSeek-Math-7b-Instruct (Shao et al., 2024); (4) Generalist Models: Claude Opus 4.1, Claude Sonnet 4.1 and Gemma-3-27b-it (Google et al., 2025).

We keep the basic tools (web search, local search and

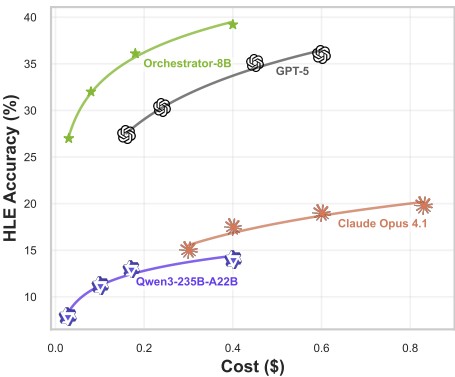

*Figure 5.* The relationship between performance and cost. Compared to strong monolithic LLM systems, Orchestrator (ours) achieves the best cost-effectiveness.

*Table 2.* Generalization performance of Orchestrator-8B on HLE, Frames and $\tau^2$-Bench.

| Model(s) | HLE ($\uparrow$) | Frames ($\uparrow$) | $\tau^2$-Bench ($\uparrow$) | Cost ($\downarrow$) | Latency ($\downarrow$) |
|---|---|---|---|---|---|
| Qwen3-8B | 12.6 | 34.9 | 38.3 | 37.9 | 10.6 |
| Llama-Nemotron-49B | 13.9 | 32.7 | 22.9 | 53.6 | 8.3 |
| Llama-3.3-70B | 13.2 | 49.3 | 12.8 | 63.3 | 10.1 |
| Qwen3-235B-A22B | 14.7 | 63.5 | 38.7 | 87.2 | 13.8 |
| Claude Opus 4.1 | 17.8 | 53.6 | 43.4 | 102.4 | 19.5 |
| GPT-5 | 16.4 | 54.8 | 44.8 | 81.3 | 14.6 |
| Orchestrator-8B | **22.0** | **73.8** | **48.8** | **34.8** | **8.2** |

code interpreter) as the same mentioned in §4.1 and generate model descriptions following the same procedures mentioned in section §3.1. Table 2 demonstrates that Orchestrator-8B shows strong skills in using models as tools. Even provided with a set of models not seen during training, Orchestrator successfully adapts to it by understanding their skills and weaknesses from model descriptions, and consistently achieves the best performance at the lowest cost across HLE, Frames and $\tau^2$-Bench. This confirms that the diverse tool configurations during training effectively enhance the generalization capabilities of Orchestrator-8B. In Appendix S, we provide more details on the comparison of orchestration performance under different model configurations, and highlight the inherent bias of GPT-5. In Appendix H, we conduct further experiments to evaluate Orchestrator-8B on pricing configurations unseen in training.

## 7. Related Work
### 7.1. From Tool Learning to Generalist Agents
Tool learning underpins advanced reasoning in LLMs, as many tasks require external APIs, search engines, or code interpreters. Early work (Schick et al., 2023; Qin et al., 2023; Qian et al., 2024) used supervised fine-tuning (SFT) on tool-labeled data like GPT-4 generated dialogues. More recently, tool use has been framed as a sequential decision-making problem optimized by RL, with systems such as WebGPT (Nakano et al., 2021), Search-R1 (Jin et al., 2025), ToRL (Li et al., 2025b), StepTool (Yu et al., 2024),

SWiRL (Goldie et al., 2025), Nemotron-Research-Tool-N1 (Zhang et al., 2025a), and ToolRL (Qian et al., 2025a). Building on this foundation, generalist agents like deep research agents (OpenAI, 2025; Google DeepMind, 2025; Perplexity AI, 2025; Moonshot AI, 2025) autonomously discover, analyze, and synthesize across sources to produce analyst-level reports, aligning with the vision of compound AI systems (Zaharia et al., 2024; Chaudhry et al., 2025). Recent open-source frameworks like SmolAgent (Roucher et al., 2025), WebAgent (Li et al., 2025a; Wu et al., 2025; Tao et al., 2025), OWL (Hu et al., 2025), AutoAgent (Tang et al., 2025), and OAgent (Zhu et al., 2025) extend this trend toward modular, robust, and accessible systems, highlighting the broader democratization of generalist agents.

### 7.2. From Tool-Use Accuracy to Efficiency and Controllability
Beyond correctness, recent work emphasizes efficiency and controllability, aiming to reduce computational costs and better align outputs with user preferences. Prompting-based methods like Self Divide-and-Conquer (Wang et al., 2025b), Efficient Agents (Wang et al., 2025c), and SMART (Qian et al., 2025b) adaptively invoke tools or fine-tune usage, though they often depend on heavy prompt engineering or curated datasets. RL provides a more flexible alternative, where reward shaping balances accuracy, efficiency, and reliability. Advances include integrating auxiliary signals (e.g., response length, task difficulty)(Aggarwal & Welleck, 2025; Arora & Zanette, 2025; Wang et al., 2025d) and combining verifiable signals with human feedback(Peng et al., 2025). A related direction is weak-to-strong generalization (Burns et al., 2024), which explores eliciting stronger models from weaker supervision. The most relevant work, OTC (Wang et al., 2025a), improves efficiency by penalizing excessive calls while preserving accuracy. Unlike the prior work, our approach addresses the broader orchestration problem by using RL to coordinate diverse tools and models, balancing correctness, efficiency, and user preference for finer alignment and more robust deployment.

## 8. Conclusion
In this work, we presented ToolOrchestra, a method for training a small orchestration model to unify diverse tools and specialized models. By training Orchestrator end-to-end with reinforcement learning, we showed that it can learn to plan adaptive tool-use strategies guided by both outcome quality, efficiency, and human preference rewards. This enables the agent to dynamically balance performance and cost, rather than relying on static heuristics or purely supervised approaches. To aid reinforcement learning, we also contribute a complex user-agent-tool synthetic dataset ToolScale. Our experiments on challenging benchmarks demonstrate that our Orchestrator-8B attains state-of-the-

art performance while operating at significantly lower cost compared to larger models. Looking ahead, we envision more sophisticated recursive orchestrator systems to push the upper bound of intelligence but also to further enhance efficiency in solving increasingly complex agentic tasks.

## Impact Statement

This work shows that small orchestration models can effectively coordinate diverse tools and language models to achieve higher performance at significantly lower computational cost. By reducing reliance on large monolithic models, our approach has the potential to lower energy consumption and improve the accessibility of advanced AI systems. At the same time, more capable orchestration systems require careful deployment, as improper tool selection or misuse could amplify downstream risks, highlighting the importance of controllability, transparency, and responsible use.

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

## A. Pilot Study

To evaluate the effectiveness of simple prompting as a method to configure an off-the-shelf language model to act as an orchestrator, we prompted GPT-5 and Qwen3-8B with a similar setting and the same prompt template we used in Section 4, allowing them to use GPT-5, GPT-5-mini, Qwen3-32B, and Qwen2.5-Coder-32B as tools and instruct the orchestrator to achieve best results while maintaining lowest cost. We then ran the model on a set of 300 HLE problems with max_tokens=32,000 and temperature T=0 and monitored the average number of times each model referred to one of its model choices. The results are shown in Figure 2. When Qwen3-8B serves as the orchestrator, it exhibits a strong tendency to delegate the task to GPT-5 (73% of the cases). We refer to this phenomenon as self-enhancement bias (Zheng et al., 2023), where the orchestrator favors its variants. In contrast, when GPT-5 serves as the orchestrator, it prefers to call GPT-5 or GPT-5-mini in 98% of the cases. We term this phenomenon other-enhancement bias, where the orchestrator favors stronger models regardless of cost considerations, even though humans instruct them to do so.

Such imbalanced invocation patterns highlight the limitations of using off-the-shelf language models as orchestrators by simple prompting: their decisions are heavily biased rather than balanced across available options, resulting in poor orchestration effectiveness. This observation motivates our method ToolOrchestra to train a dedicated small orchestrator to decide when and how to invoke more intelligent language models.

## B. Evaluation Benchmarks

- **Humanity's Last Exam (HLE)** (Phan et al., 2025). A large-scale benchmark comprising PhD-level questions across mathematics, humanities, natural sciences and more. It evaluates the model capabilities to perform iterative search and intensive reasoning. Questions are multiple-choice or short-answer, with 10–14% requiring images. We use the text-only subset, designed to be unambiguous and not solvable by simple web search.

- **FRAMES** (Krishna et al., 2024). A dataset for end-to-end evaluation of retrieval-augmented generation (RAG), covering factuality, retrieval accuracy, and reasoning. It contains 824 multi-hop questions requiring 2–15 Wikipedia articles, spanning numerical, tabular, temporal, and multi-constraint reasoning.

- $\tau^2$**-Bench** (Barres et al., 2025). A benchmark to evaluate model capabilities to use tools and solve problems in conversations with users. It includes tasks in three domains: telecom, retail and airline.

## C. Model description for Qwen3-32B

The model shows advanced mathematical and quantitative reasoning, often solving complex problems and only faltering on highly specialized or computationally heavy items. Scientific domain knowledge is strong—especially in biology—with solid performance in physics and engineering; chemistry is mixed, with notable weaknesses in exact nomenclature and InChI outputs. Logical problem-solving is high, while humanities knowledge is moderate and uneven, with gaps in niche scholarly details. Coding and function call abilities are moderate, where it makes mistakes in parameters from time to time. Overall, the model has broad knowledge and robust analytic skills, but accuracy drops on narrow, recent, or rote-precision tasks, particularly in chemical informatics.

## D. Tools in training

Below is the complete list of tools used in the training. For each example rollout, we randomly sample a subset of them to simulate heterogeneous availability of tools:

- Query writer: GPT-5 (OpenAI, b), GPT-5-mini (OpenAI, b), meta-llama/Llama-3.3-70B-Instruct (Dubey et al., 2024), meta-llama/Llama-3.1-8B-Instruct (Dubey et al., 2024), deepseek-ai/DeepSeek-R1 (Guo et al., 2025), nvidia/Llama-3.1-Nemotron-Ultra-253B-v1 (Bercovich et al., 2025), microsoft/Phi-4-mini-instruct (Abouelenin et al., 2025), google/gemma-3-27b-it (Google et al., 2025), Qwen/Qwen3-32B (Yang et al., 2025)

- Web search: We use Tavily search API [5] to provide orchestrator real-time web access.

- Local search: Faiss index with Qwen/Qwen3-Embedding-8B (Zhang et al., 2025b)

---

[5] https://www.tavily.com/

- Code writer + interpreter: We use GPT-5 (OpenAI, b), GPT-5-mini (OpenAI, b), bigcode/starcoder2-15b (Lozhkov et al., 2024), and Qwen/Qwen2.5-Coder-32B-Instruct (Hui et al., 2024) as code expert models to write code. We also implemented a Python sandbox to execute the code.

- Math models: Qwen/Qwen2.5-Math-72B (Yang et al., 2024), Qwen/Qwen2.5-Math-7B (Yang et al., 2024)

- Generalist models: GPT-5 (OpenAI, b), GPT-5-mini (OpenAI, b), meta-llama/Llama-3.3-70B-Instruct (Dubey et al., 2024), meta-llama/Llama-3.1-8B-Instruct (Dubey et al., 2024), deepseek-ai/DeepSeek-R1 (Guo et al., 2025), nvidia/Llama-3_1-Nemotron-Ultra-253B-v1 (Bercovich et al., 2025), microsoft/Phi-4-mini-instruct (Abouelenin et al., 2025), Qwen/Qwen3-32B (Yang et al., 2025)

## E. Third-party API

Here is a list of third-party APIs. We used pricing configurations for training:

- TogetherAI: https://www.together.ai/

- Venice AI: https://docs.venice.ai/overview/about-venice

- Chutes: https://chutes.ai/

- NEBIUS: https://nebius.com/

- Lambda: https://lambda.ai/

- Hyperbolic: https://docs.hyperbolic.xyz/docs/welcome-to-hyperbolic

- Cloudflare: https://developers.cloudflare.com/

- Novita: https://novita.ai/

- AIML: https://aimlapi.com/

- Fireworks AI: https://fireworks.ai/

In the evaluation, we apply the pricing from Together AI for fair comparison.

## F. Humane preference example

**Tools**; $T$ = [ Web search, local search, Qwen/Qwen3-235B-A22B, meta-llama/Llama-3.3-70B-Instruct, o3-mini, o3 ]
Preference instruction: $PI$ = I am a company employee and there is some confidential information in my server. There are many GPUs in the server, so I can host open-sourced models or retrievers. It would be great if we can avoid API calling as much as possible.
Preference vector: $P$ = [0,1,1,1,0,0,0,0,0] Explanation: The first digit in the preference vector corresponds to the first tool in $T$; The second digit in the preference vector corresponds to the second tool in $T$, etc. The last three digits in $P$ corresponds to accuracy, cost and latency, aligned with the definitions in §3.2. Therefore, this preference vector means the user prefers to use local search, Qwen/Qwen3-235B-A22B, meta-llama/Llama-3.3-70B-Instruct.

## G. Use of LLMs Disclosure

We used GPT-5 to polish the writing, primarily in the Abstract, Introduction, Methodology, and Experiments sections, to improve the grammar, clarity, and readability. The research ideas, methodology, experiments, and analyses were entirely conducted by the authors.

*Table 3.* Generalization performance under different a pricing configuration. Orchestrator-8B consistently performs the best in terms of performance, cost and latency, which illustates the robustness of the Orchestrator

|  | HLE ($\uparrow$) | Frames ($\uparrow$) | $\tau^2$-Bench ($\uparrow$) | Cost ($\downarrow$) | Latency ($\downarrow$) |
|---|---|---|---|---|---|
| Qwen3-8B | 29.7 | 68.2 | 71.9 | 27.4 | 17.9 |
| Nemotron-49B | 25.6 | 57.8 | 66.3 | 24.3 | 17.2 |
| Llama-3.3-70B | 19.6 | 52.2 | 55.4 | 17.9 | 12.0 |
| Qwen3-235B-A22B | 32.4 | 74.1 | 75.3 | 27.9 | 20.8 |
| Claude Opus 4.1 | 34.5 | 72.3 | 76.4 | 52.3 | 25.1 |
| GPT-5 | 20.8 | 57.3 | 61.9 | 17.5 | 13.2 |
| **Orchestrator-8B** | **36.9** | **76.6** | **80.4** | **7.5** | **7.8** |

# H. Generalization of pricing configurations

In Section 6.3, we examined Orchestrator-8B's ability to generalize to unseen tools. Here, we investigate its generalization across heterogeneous pricing regimes, where the same tools are assigned different costs. We evaluate whether the model adapts by adjusting its tool-calling strategy to optimize outcomes, efficiency, and alignment with user preferences—reflecting realistic settings in which tool costs vary across users. We test Orchestrator-8B under a pricing configuration not encountered during training. Specifically, we use the pricing configuration from deepinfra[6]. As shown in Table 3, it consistently delivers superior performance, cost efficiency, and accuracy. These results demonstrate that training with diverse pricing configurations produces a model that is not constrained to a particular tool setup and can robustly generalize across diverse user scenarios.

# I. Data Synthesis

**ToolScale.** To enable end-to-end RL training of Orchestrator, we require data involving user-agent-tool interaction trajectories, but such verifiable data is scarce. To generate such high-quality data, we devise a two-step process: (1) simulating rich user-agent-tool environments, including creating database schemas and tool APIs; and (2) based on the environment, generating diverse user tasks together with their corresponding ground truth solutions. We further ensure quality by carefully verifying that each task is solvable using the provided databases and tool APIs. Figure 3 provided an overview of this process. Firstly, to simulate real-world user-agent-tool environments scalably, we choose a domain $D$ and then ask an LLM to generate a database which includes schema, major subjects to focus on and database entries (as illustrated in the top-left of Figure 3). Each entry is then checked to ensure coherence, adherence to the schema, and consistency across content, logic, and entities. Based on the given domain $D$, LLM proposes frequently-used tools. Secondly, to increase the diversity of the task instructions, LLM first proposes diverse intents frequently seen in domain $D$, which are later converted to specific tasks based on detailed database information. Each generated task consists of task instruction $I$, gold function calls $A = a_1, a_2, ..., a_l$, and short information $o$ that must be mentioned during the process to solve the task. To enhance the difficulty of the generated tasks, we leverage an additional LLM to complicate tasks by adding more complexities such as more constraints.

To ensure the quality of the synthesized data, we filter the data to remove a task if: (1) the execution of golden function calls reports an error; (2) LLMs cannot solve it in pass@8; and (3) the task can be solved without any actions. In Appendix J, we list the statistics of the generated data in each domain. More examples and prompts used to synthesize data can be found in Appendix K. To evaluate whether a trajectory $\tau$ solves the given task, we define the following criteria: (1) *execution correctness*, namely whether the database content matches after executing the golden function calls $A$ and the trajectory $\tau$; (2) *process fidelity*, i.e., whether the predefined information $o$, which is required to be communicated in the process to solve the task, is mentioned in $\tau$; (3) *operation completeness*, that is whether the database entries operated in the ground truth trajectory $A$ are also operated in $\tau$. We consider $\tau$ solves the task if all of three criteria are satisfied, or fails otherwise.

# J. Breakdown of ToolScale

We show the breakdown of tools, DB entries, and tasks in each domain in Table 4.

# K. Data synthesis prompts and examples

We show the data synthesis prompts and examples from Table 5 to 13.

---

[6]https://deepinfra.com

*Table 4.* Statistics of ToolScale: the number of tools, database entries, and tasks per domain.

|  | Finanace | Sport | E-commerce | Medicine | Entertainment | Railway | Restaurant | Education | Travel | Weather |
|---|---|---|---|---|---|---|---|---|---|---|
| Tools | 22 | 19 | 15 | 19 | 24 | 25 | 23 | 21 | 15 | 14 |
| DB Entries | 686 | 423 | 577 | 920 | 852 | 790 | 683 | 816 | 752 | 549 |
| Tasks | 396 | 247 | 343 | 622 | 561 | 414 | 348 | 426 | 331 | 375 |

*Table 5.* Model prompts to generate subjects in a domain

Generate a list of major subjects in {domain}.
Output using the following format:
```

[subject1, subject2, ...]
```

## L. Calculation of rewards for preference-aware benchmark

During training, we directly follow the Equation 2 to calculate rewards. In the evaluation, we use the following procedure. Following the definition in §3.2, we have a tool set $\{t_1, t_2, ..., t_n\}$ and a rollout trajectory $\tau$, we consider the vector $M^\tau = [m^\tau_{t_1}, m^\tau_{t_2}, \ldots, m^\tau_{t_n}, r^\tau_{\text{outcome}}, r^\tau_{\text{compute}}, r^\tau_{\text{latency}}]$, where $m^\tau_{t_\bullet}$ is the number of times tool $t_\bullet$ is invoked in $\tau$. In the evaluation, we obtain the baseline vector $M_0$ by running the starting checkpoint, e.g., Qwen3-8B. For example, if we would like to evaluate a checkpoint $CKPT_s$ that is trained for $s$ steps from a base Qwen3-8B model, then we first run Qwen3-8B on the benchmark and record the vector $M_0^{\tau(e)}$ as the baseline vector for the Qwen3-8B's trajectory $\tau(e)$ for each example $e$ of the benchmark. We then obtain $M_s^{\tau(e)}$ by running $CKPT_s$ on the same example $e$. $M_s^{\tau(e)}$ is normalized as

$$M^{\tau(e)}_{\text{normalized},s}[k] = \begin{cases} M_s^{\tau(e)}[k]/max(1, M_0^{\tau(e)}[k]) & \text{if } 1 \leq k \leq n+1 \\ M_0^{\tau(e)}[k]/max(1, M_s^{\tau(e)}[k]) & \text{otherwise.} \end{cases} \tag{5}$$

We then proceed to calculate the final preference-aware reward for the example $e$ as:

$$R^e(\tau) = \begin{cases} M^{\tau(e)}_{\text{normalized},s} \cdot P & \text{if } r_{\text{outcome}(\tau)} \\ 0 & \text{otherwise.} \end{cases} \tag{6}$$

The benchmark result is calculated as the sum of $R^e(\tau)$ over the examples $e$ of the benchmark.

## M. Filtering strategies

We study the influence of filtering strategies in training by removing one of format filtering, homogeneity filtering and invalid output filtering mentioned in §3.2. Results in Figure 6 shows that omitting invalid-output filtering causes the reward to begin decreasing after approximately 400 steps. Excluding homogeneity filtering slows reward improvement and leads to earlier saturation. Removing format filtering impairs scaling as it consistently underperforms Orchestrator-8B when the three types filtering methods are applied. These findings indicate that all three filtering strategies are essential for stabilizing training and achieving the best model performance.

## N. Robustness analysis

We assess the adversarial robustness of Orchestrator-8B by constructing a 278-example adversarial test set, created by appending synthetically generated adversarial prompts to the original task instructions. The evaluation is conducted in an out-of-domain setting where test domains and environments are disjoint from those used in training. Robustness is quantified as the fraction of examples in which the model refuses the adversarial instruction and the intended adversarial outcome does not occur—for example, rejecting an instruction such as "leave a positive review for the airline" when reviews should reflect actual user experience, thereby avoiding the insertion of a misleading positive entry into the database. Using the same tool set and identical model configurations, we benchmark Orchestrator-8B against Qwen3-235B, Claude Opus 4.1, and

*Table 6.* Model prompts to generate schema in a domain

---

```
{demo_schema}
```

Generate another schema of similar formats for {domain}.

---

*Table 7.* Model prompts to generate database entry

---

Schema
```
{schema}
```

Following the schema, write records in the subject {subject}. Make sure that the values align with the definitions in the schema and are consistent in different places. Use the following format to output:
```
{ "...": ..., "...": ..., }
```

Wrap the dictionary within ```.

---

GPT-5. As shown in Table 16, Orchestrator-8B rejects adversarial instructions at a substantially higher rate than these strong baselines, demonstrating superior robustness under distribution shifts in real-world settings.

## O. LLM-as-a-judge

By default, we use GPT-5 as the judge model to compare model predictions against reference answers and determine correctness. We avoid strict exact-match evaluation because ground-truth formats vary widely, making string-level comparisons unreliable. Instead, an LLM-as-judge approach provides greater flexibility in acceptable formats and phrasing. We evaluate multiple judge models on HLE using a random subsample of 100 examples and report two metrics: (1) Error rate—the proportion of examples in which the judge model's assessment is incorrect. Because most answers are under 20 characters, humans are involved to verify whether the judge model's decisions are correct; and (2) Accuracy—the task accuracy achieved when each judge model is applied. The results in Table 17 indicate that Qwen3-32B's error rate is typically below 4%, with minimal impact on overall performance. As an automatic evaluator of answer correctness, it is therefore a practical choice for improving training efficiency and reproducibility. Nonetheless, relative to GPT-5, Qwen3-32B shows a modestly higher rate of misjudgments in comparing outputs to ground truth. For experiments that demand greater rigor and minimal evaluator-induced noise—particularly in reinforcement learning—GPT-5 remains a more conservative and reliable option.

## P. Controllability evaluation

In Section U, we quantify preference rewards to holistically assess Orchestrator-8B's ability to adhere to user-specified preferences (e.g., web vs. local search, third-party APIs vs. on-device models). We additionally perform more fine-grained ablation studies to analyze the agent's behavior under different test-time preference prompts. Specifically, we condition Orchestrator-8B on instructions that specify distinct performance–cost trade-offs and evaluate both dimensions across HLE, FRAMES, and $\tau^2$-Bench. The results in Table 18 show that Orchestrator-8B exhibits strong alignment with user instructions when balancing performance and cost, and that modifying the preference instructions at inference time reliably shifts the agent's behavior to complete tasks accordingly.

## Q. Size of orchestrator

Our vision is to employ small models as tool-orchestrating agents for complex tasks. In this context, an 8B-parameter model is relatively small—roughly two orders of magnitude fewer parameters than frontier models (e.g., $> 100B$)—and

*Table 8.* Model prompts to validate database entry

Schema
```
{schema}
```

Database entry
```
{db_entry}
```

Please check whether the following conditions are satisfied:
Condition 1. The database entry strictly aligns with the fields and type definitions in the schema.
Condition 2. The values in the database entry are consistent across different places, e.g., id, name, etc.
Condition 3. The database content is logical, natural, and reasonable.
Output using the following format:
```
{ "condition 1": "satisfied or not satisfied, "condition 2": "satisfied or not satisfied, "condition 3": "satisfied or not satisfied, }
```

*Table 9.* Model prompts to generate functions

Demonstration function
```
{demo_function}
```

Following the formats of demonstration function, write frequently-used functions in {domain}.
Wrap the functions within ```.

substantially more efficient at inference. We consider the 8B experiments a first step in a broader family of orchestrators spanning 1B–8B parameters. Using the same training pipeline, we also trained 1.7B and 4B variants and report results in Table 19. The 4B orchestrator delivers strong results that closely approach the 8B on HLE, FRAMES, and tau2-bench, while preserving clear efficiency advantages over larger models. With larger sizes of orchestrator models, they exhibit stronger reasoning and planning capabilities, which leads to better orchestration of tools and models, and improved performance and efficiency.

## R. Data example

Below is an example of difficult tasks synthesized in the restaurant domain.

Question: Today is July 23, 2025. On the next Wednesday, Jasper would like to book the same dinner to welcome another group of guests as he did on the last Monday. If both dishes and the table (that can serve the same number of people) are available, use the same payment method to pay and record points to his membership card.

One of the gold trajectories to finish the task goes through the following stages: (1). Use $look\_calendar$ to find the dates of last Monday and next Wednesday; (2). Use $get\_order$ to find the order of Jasper last Monday, which includes the information of the dishes S, the number of people N and the payment method P; (3). Use $find\_table$ to find all the tables T that could serve N people; (4). For each table in T, use $check\_table\_availability$ to check their availability for next

*Table 10.* Model prompts to generate intents

| |
|---|
| Functions |
| ``` |
| {functions} |
| ``` |
| |
| |
| Propose realistic intents in {domain} that could be solved by the functions above. Use the following format to output: |
| ```. |
| [ |
| "purpose 1", |
| "purpose 2", |
| ... |
| ] |
| ```. |

*Table 11.* Model prompts to generate tasks

| |
|---|
| Functions |
| ``` |
| {functions} |
| ``` |
| Database |
| ``` |
| {database} |
| ``` |
| |
| |
| Propose a realistic task with the intent: {intent}. Use the following format to output: |
| ```. |
| {task_template} |
| ```. |

Monday dinner; (5). For each dish in S, use $find\_chef$ to find all the chefs C that can cook it, and for each chef in C, use $check\_chef\_on\_duty$ to check whether they are on duty for the next Monday dinner; (6). If a table to serve N people is found available and a chef is found on duty for every dish, then use $get\_dish$ to obtain the price of each dish; (7), Use calculator to sum the price of all dishes; (8). Use $make\_order$ to book the table with all dishes; (9). Use $pay\_bill$ to make the payment with the payment method P; (10). Use $get\_membership$ to get Jasper's membership ID; (11). Use $calculate\_points$ to calculate the points for the order; (12). Use $update\_points$ to update the points in Jasper's account.

## S. Orchestration performance with different model pools

Tables 1 and 2 present the orchestration performance under different model pool configurations. The results show that Orchestrator-8B consistently achieves the strongest performance across settings, demonstrating its ability to generalize and to select appropriate models beyond its training configurations. A consistent performance degradation is observed when moving from Table 1 to Table 2. This decline can be attributed to the more capable tool sets employed in Table 1, which provide richer knowledge and stronger reasoning abilities on the evaluated benchmarks. These findings indicate that end-task performance depends not only on effective orchestration among heterogeneous models and tools, but also on the intrinsic capabilities and domain expertise of the underlying components.

Furthermore, as shown in Table 1, GPT-5 exhibits an inherent selection bias and frequently defaults to GPT-5-mini (see Section 6.1), resulting in inferior performance. To further investigate this behavior, we analyze the model invocation distributions of GPT-5 and Orchestrator-8B in the setting of Table 2, as reported in Table 20. The results indicate that GPT-5

*Table 12.* Model prompts to evolve tasks

Functions
```
{functions}
```

Database
```
{database}
```

Old task: {task}

Make a new task by adding more complexity to the old task. You can add constraints, involve more entities, make the situation more tricky, etc. Use the following format to output:
```.
{task_template}
```.

predominantly relies on the strongest available model, Claude Opus 4.1, in this setting, while heavily favoring GPT-5-mini in the configuration of Table 1, revealing instability in its model-selection strategy. In contrast, Orchestrator-8B maintains a more balanced utilization of diverse models across both experimental settings, highlighting its robustness and superior generalization capability.

## T. Reasoning effort

We conduct experiments to investigate the influence of reasoning efforts on the performance of the orchestrator model. We specify reasoning efforts in the parameters for Claude Opus 4.1 and GPT-5, and in the prompt for the Orchestrator-8B. We use the tool set to include basic tools, specialized LLMs and generalist LLMs. The number of reasoning tokens is averaged across three benchmarks. The results in Table 22 show that the performance improves as the model uses more tokens in reasoning, which indicates that the orchestration benefits from scaling up the model compute.

## U. User Preferences

To assess Orchestrator-8B's ability to adapt to heterogeneous user preferences at test time, we evaluate it on the Preference-aware test set described in §3.3, where we concatenate each question with an additional preference instruction. We evaluate the model preference adherence performance by calculating the preference-aware rewards defined in Appendix L. Table 21 shows that, even strong monolithic systems such as GPT-5 struggle to faithfully follow user preferences. In contrast, Orchestrator-8B exhibits remarkably better adherence to user preferences.

*Table 13.* Database schema example

```
{
"movies": {
"MMMMMMM": { movie_id
"movie_id": "MMMMMMM",
"title": "...",
"genres": ["Action", "Adventure", "Comedy", "Drama", "Horror", "Thriller", "Romance",
"Science Fiction", "Fantasy", "Mystery"],
"runtime_minutes": ...,
"mpaa_rating": "...",
"languages_audio": ["..."],
"subtitles": ["..."],
"formats": ["2D", "3D", "IMAX", "Dolby"],
"release_date": "YY-MM-DD",
"end_of_run_est": "YY-MM-DD",
"cast": [
{ "name": "...", "role": "..." }
],
"crew": {
"director": "...",
"writer": "...",
"producer": "...",
"music": "..."
},
"synopsis": "..."
},
...
},
...
}
```

*Table 14.* The average number of calls on each tool when various models serve as the orchestrator to solve an instance (averaged across HLE, Frames and $\tau^2$-bench). Qwen-32B refers to Qwen/Qwen3-32B (Yang et al., 2025), Coder-32B refers to Qwen/Qwen2.5-Coder-32B-Instruct (Hui et al., 2024), Math-7B refers to https://huggingface.co/Qwen/Qwen2.5-Math-7B-Instruct (Yang et al., 2024), Math-72B refers Qwen/Qwen2.5-Math-72B-Instruct (Yang et al., 2024), and Llama-70B refers to meta-llama/Llama-3.3-70B-Instruct (Dubey et al., 2024). Compared to other strong foundation models, Orchestrator-8B achieves better results (Table 1) while making few calls to GPT-5.

| Model | GPT-5 | GPT-5-mini | Qwen-32B | Coder-32B | Math-72B | Math-7B | Llama-70B | Local search | Web search | Code interpreter |
|---|---|---|---|---|---|---|---|---|---|---|
| Qwen3-8B | 6.0 | 0.5 | 1.4 | 0.5 | 0.0 | 0.0 | 0.0 | 0.8 | 1.2 | 1.6 |
| Nemontron-49B | 5.1 | 1.6 | 0.5 | 0.8 | 0.1 | 0.1 | 0.3 | 0.7 | 0.9 | 1.4 |
| Llama-3.3-70B | 1.8 | 0.0 | 0.1 | 0.0 | 1.4 | 0.3 | 4.8 | 0.6 | 1.4 | 1.3 |
| Qwen3-235B-A22B | 6.2 | 0.3 | 0.6 | 1.2 | 0.6 | 0.1 | 1.1 | 1.4 | 1.0 | 2.2 |
| Claude Opus 4.1 | 6.2 | 0.2 | 0.3 | 0.3 | 0.1 | 0.0 | 0.1 | 1.0 | 1.3 | 1.4 |
| GPT-5 | 2.7 | 5.6 | 0.0 | 0.2 | 0.0 | 0.0 | 0.0 | 0.5 | 0.7 | 1.0 |
| Orchestrator-8B | 1.6 | 1.7 | 1.3 | 0.2 | 0.0 | 0.1 | 0.0 | 1.8 | 0.7 | 0.8 |

*Table 15.* The cost and latency of LLMs in $\tau^2$-Bench. Orchestrator-8B consistently shows better performance with lower cost and latency.

| Tools | Model(s) | $\tau^2$-Bench ($\uparrow$) | Cost ($\downarrow$) | Latency ($\downarrow$) |
|---|---|---|---|---|
| Basic tools | Qwen3-8B | 40.7 | 1.6 | 2.3 |
| | Llama-Nemotron-49B | 23.2 | 2.7 | 3.6 |
| | Llama-3.3-70B | 17.6 | 3.1 | 4.5 |
| | Qwen3-235B-A22B | 52.9 | 12.6 | 10.6 |
| | Claude Opus 4.1 | 46.0 | 81.2 | 32.8 |
| | GPT-5 | 77.7 | 31.3 | 20.2 |
| Basic tools, Specialized LLMs Generalist LLMs | Qwen3-8B | 72.3 | 27.9 | 18.4 |
| | Llama-Nemotron-49B | 66.7 | 25.8 | 17.5 |
| | Llama-3.3-70B | 55.8 | 20.1 | 14.2 |
| | Qwen3-235B-A22B | 75.6 | 30.0 | 22.6 |
| | Claude Opus 4.1 | 76.8 | 52.8 | 24.1 |
| | GPT-5 | 62.3 | 18.2 | 14.5 |
| | **Orchestrator-8B** | **80.2** | **10.3** | **8.6** |

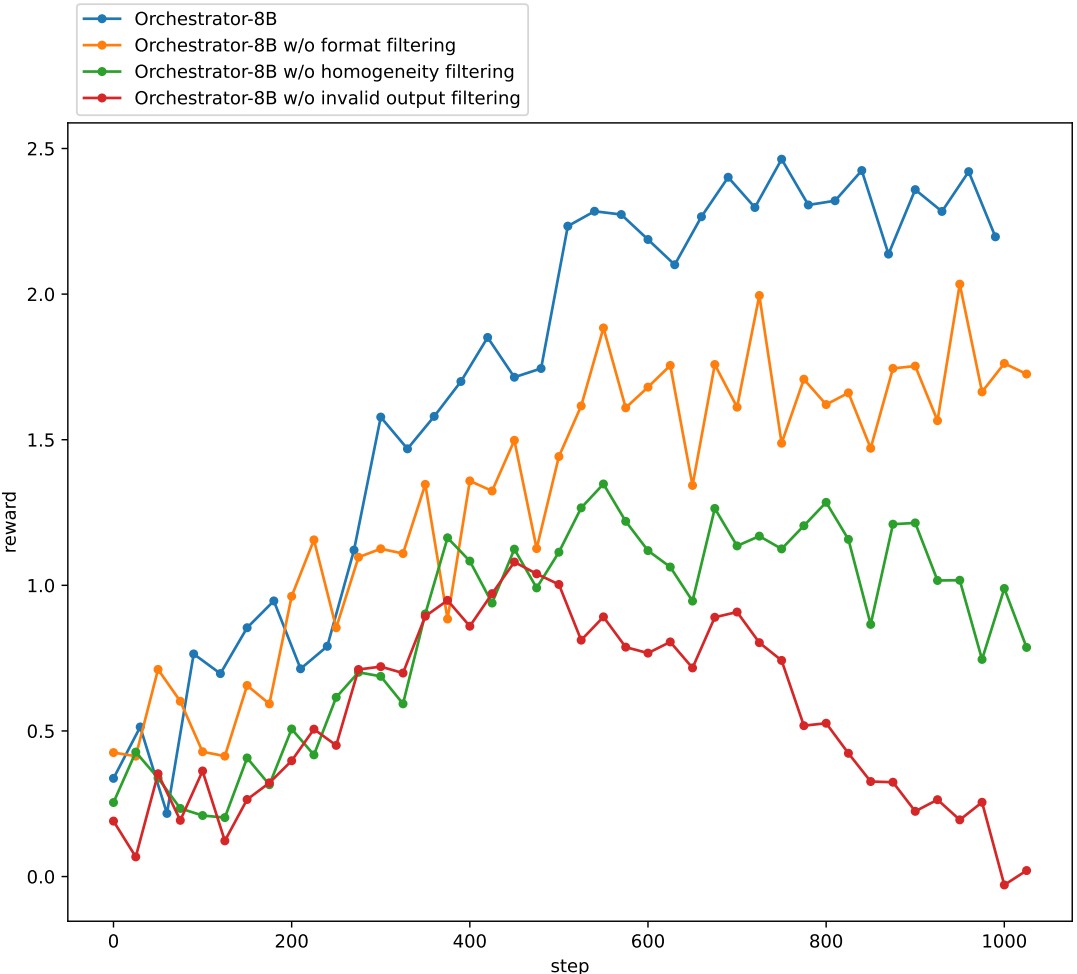

*Figure 6.* Rewards of Orchestrator-8B with different filering strategies.

*Table 16.* The percentage of examples where adversarial instructions are not followed when different models are used as the orchestrator. The results show that Orchestrator-8B achieves the best results, indicating that it is safe and robust to adversarial attack under distribution shift.

| Qwen3-235B-A22B | Claude Opus 4.1 | GPT-5 | Orchestrator-8B |
|---|---|---|---|
| 47.6 | 52.3 | 58.7 | 65.4 |

*Table 17.* The error rate of judge model and its influence on task accuracy. The results show that the error rate of Qwen3-32B is slightly higher than GPT-5, but is generally below 5% and has only minimal influence on overall task accuracy.

| Judge model | Model(s) | Accuracy | Error rate |
|---|---|---|---|
| Qwen3-32B | Qwen3-8B | 31.8 | 2.7 |
| | Llama-Nemotron-49B | 23.6 | 3.3 |
| | Llama-3.3-70B | 17.5 | 3.2 |
| | Qwen3-235B-A22B | 31.9 | 2.8 |
| | Claude Opus 4.1 | 33.7 | 3.2 |
| | GPT-5 | 22.6 | 3.4 |
| | Orchestrator-8B | 36.5 | 2.3 |
| GPT-5 | Qwen3-8B | 30.6 | 1.4 |
| | Llama-Nemotron-49B | 25.8 | 1.7 |
| | Llama-3.3-70B | 19.7 | 2.1 |
| | Qwen3-235B-A22B | 32.8 | 1.8 |
| | Claude Opus 4.1 | 34.6 | 1.0 |
| | GPT-5 | 21.2 | 1.5 |
| | Orchestrator-8B | 37.1 | 1.2 |

*Table 18.* The performance of Orchestrator-8B under different preference instructions on the trade-off between accuracy and cost. The results show the strong instruction-following capability of Orchestrator-8B and that users can control the balance of accuracy and cost by altering the prompt.

| Preference instruction | HLE | FRAMES | $\tau^2$-Bench | Cost |
|---|---|---|---|---|
| Exclusively optimize the performance without considering the cost | 39.2 | 79.8 | 82.4 | 25.7 |
| Top-tier performance, but occasionally reducing the cost when possible | 37.7 | 77.5 | 80.8 | 16.4 |
| Good performance, but need to balance the cost and reduce the usage of expensive models and tools | 37.1 | 76.3 | 80.2 | 9.2 |
| Reasonable performance under limited budgets | 30.6 | 71.5 | 74.9 | 4.6 |

*Table 19.* The performance of Orchestrator in various sizes. The results show that both the performance and efficiency improve when larger models are used as the orchestrator.

| Model(s) | HLE | FRAMES | $\tau^2$-Bench | Cost | Latency |
|---|---|---|---|---|---|
| Qwen3-8B | 30.6 | 68.9 | 72.3 | 27.6 | 18.3 |
| Llama-Nemotron-49B | 25.8 | 57.9 | 66.7 | 25.6 | 17.1 |
| Llama-3.3-70B | 19.7 | 52.4 | 55.8 | 19.7 | 13.4 |
| Qwen3-235B-A22B | 32.8 | 74.2 | 75.6 | 29.7 | 21.2 |
| Claude Opus 4.1 | 34.6 | 72.8 | 76.8 | 52.5 | 25.6 |
| GPT-5 | 21.2 | 57.5 | 62.3 | 17.8 | 13.6 |
| Orchestrator-1.7B | 32.3 | 72.4 | 75.8 | 12.6 | 10.8 |
| Orchestrator-4B | 35.6 | 74.8 | 78.4 | 11.4 | 9.3 |
| Orchestrator-8B | 37.1 | 76.3 | 80.2 | 9.2 | 8.2 |

*Table 20.* The percentage of each model called by GPT-5 and Orchestrator-8B. The results show that GPT-5 has strong bias to call the strongest model among others.

| Orchestrator | Claude Opus 4.1 | Claude sonnet 4.1 | o3-mini | GPT-4o | Codestral-22B-v0.1 | OpenMath-Llama-2-70b | DeepSeek-Math-7b-instruct | Gemma-3-27b-it |
|---|---|---|---|---|---|---|---|---|
| GPT-5 | 65.6 | 25.4 | 1.7 | 4.3 | 0.1 | 1.4 | 0.3 | 1.2 |
| Orchestrator-8B | 18.7 | 19.2 | 14.8 | 15.7 | 16.9 | 1.2 | 1.9 | 11.6 |

*Table 21.* Preference performance comparison. The results show that Orchestrator-8B best adapts to user preference during test time.

| Model(s) | HLE | Frames | $\tau^2$-Bench |
|---|---|---|---|
| Qwen3-8B | 25.3 | 43.2 | 55.7 |
| Llama-Nemotron-49B | 27.6 | 50.1 | 56.9 |
| Llama-3.3-70B | 22.3 | 44.5 | 55.4 |
| Qwen3-235B-A22B | 37.9 | 54.5 | 68.2 |
| Claude Opus 4.1 | 40.2 | 63.4 | 73.5 |
| GPT-5 | 34.6 | 62.3 | 70.3 |
| Orchestrator-8B | **46.7** | **68.4** | **79.5** |

*Table 22.* Model performance under various reasoning efforts. The results show superior performance as reasoning tokens increase.

| Reasoning effort | Model | HLE | FRAMES | $\tau^2$-Bench | Reasoning tokens |
|---|---|---|---|---|---|
| Low | Claude Opus 4.1 | 32.4 | 70.4 | 74.7 | 19,128 |
| | GPT-5 | 17.7 | 55.8 | 59.8 | 17,536 |
| | Orchestrator-8B | 32.8 | 71.6 | 75.4 | 12,382 |
| Medium | Claude Opus 4.1 | 34.6 | 72.8 | 76.8 | 25,727 |
| | GPT-5 | 21.2 | 57.5 | 62.3 | 23,568 |
| | Orchestrator-8B | 37.1 | 76.3 | 80.2 | 21,324 |
| High | Claude Opus 4.1 | 34.9 | 73.6 | 77.5 | 31,236 |
| | GPT-5 | 22.1 | 58.7 | 62.6 | 29,783 |
| | Orchestrator-8B | 38.9 | 79.6 | 82.5 | 26,382 |

