# OpenReview forum: "ToolOrchestra: Elevating Intelligence via Efficient Model and Tool Orchestration"
_ICML.cc/2026/Conference — ICML 2026 regular_

### Official Review · Reviewer_KSvJ · 2026-03-11

**Soundness:** 4
**Presentation:** 4
**Significance:** 3
**Originality:** 3
**Overall Recommendation:** 6
**Confidence:** 4

**Summary:**

ToolOrchestra proposes a small orchestrator for a larger system of tools and models. The paper trains an 8B model to delegate queries -when to call external tools, and when to delegate parts of the problem to stronger or more specialized LLMs. Then they train the orchestrator end-to-end with reinforcement learning so it learns multi-step reasoning and tool-use trajectories. The reinforcement learning objective has three parts. First is an outcome reward based on whether the task is solved. Second are efficiency rewards that penalize high monetary cost and long wall-clock latency. Third is a preference component meant to reflect user choices, such as favoring cheaper, faster, or privacy-preserving behavior. So the orchestrator is trained to make trade-offs, not merely maximize raw benchmark score.

**Compliance With Llm Reviewing Policy:**

Affirmed.

**Key Questions For Authors:**

Same as weakness

**Limitations:**

Same as weakness

**Strengths And Weaknesses:**

Strength -
1. The central idea of the paper and design choices are novel specially the RL training optimizing for compute cost, latency, and user preferance identifying failures in prompt-only optimization
2. Evaluation across hard benchmarks are covered well
3. Clear writing
4. synthetic data pipeline for creating verifiable multi-turn tool-use tasks is a practical contribution on its own.

Weakness -
1. Lack of statistical rigor
2. Reward not motivated

---

> ### Author Rebuttal · Authors · 2026-03-31
>
> We thank the reviewer for their thoughtful and constructive comments. We are pleased that the reviewer considers our approach to optimizing compute cost, latency, and user preference to be novel, views the evaluation as thorough, finds the writing to be well-presented, and regards the synthetic data pipeline as a useful contribution. Below, we address the questions raised in the review.
>
> **HLE comparison**
> We appreciate the reviewer raising the question of statistical rigor. To clarify, the baselines in the first section (Existing baselines) of Table 1 serve as general reference points drawn from results reported across the wider research community and industry, while all of our own experiments focus exclusively on the text-only subset. For a more rigorous evaluation, we ran experiments across three distinct settings: no tools, basic tools, and basic tools combined with specialized and generalist LLMs. In terms of HLE performance, GPT-5 scores 35.2% on the full dataset, and our own reproduction yields 35.1% on the text-only subset. Importantly, all our in-house evaluations are performed on the same text-only subset under a controlled setting, ensuring that all comparisons are fair and directly comparable.
>
> **Ablation study on reward component**
> We thank the reviewer for the insightful questions regarding our reward design. As described in Section 3.2 (Lines 180–185), the weights assigned to different reward signals are governed by the preference vector P, which reflects the degree to which a user wishes to prioritize a given objective.
>
> We concur with the reviewer that detailed ablation studies would provide greater clarity on how the multi-objective reward drives improvements during training. Following this suggestion, we conducted additional experiments under the following configurations: (1) outcome-only reward, (2) outcome + efficiency, (3) outcome + preference for API-based tools (simulating users who prefer API-based tools), and (4) outcome + preference for locally deployed Qwen-series models (simulating scenarios in which users have easy access to local Qwen-series models over other tools). To better capture the effect of preference-related rewards, we also include three supplementary metrics: the average proportion of API-based tool calls per trajectory, the average proportion of Qwen-series model calls per trajectory, and the preference-aware reward score (measuring how faithfully the model adheres to user preferences), averaged across the three benchmarks defined in Appendix L.
>
> | Reward | HLE  | FRAMES | $\tau^2$-Bench | Cost | Latency | Percentage of calls to API-based tools | Percentage of calls to Qwen-series models |  Preference-aware rewards    |
> | -------- | ---- | ------ | --------------- | ---- | ------- | -------------------------------------- | ----------------------------------------- | -- |
> | outcome-only  | 39.2 | 80.1   | 82.4            | 13.2 | 11.6    | 59.7                                   | 11.2      |        41.1                                |
> | outcome+efficiency                                          | 36.7 | 76.4   | 79.8            | 8.8  | 7.9     | 46.6                                   | 20.3      |         41.6                               |
> | outcome+preference over API-based tools                     | 39.6 | 79.2   | 81.8            | 13.9 | 12.2    | 66.3                                   | 5.2       |    40.8                               |
> | outcome+preference over locally deployed Qwen-series models | 35.3 | 73.7   | 74.6            | 7.2  | 6.3     | 34.8                                   | 36.8      |    40.2                                |
> | Multi-objective (settings in paper)                      | 37.1 | 76.3   | 80.2            | 9.2  | 8.2     | 48.7                                   | 19.5    |     64.9                                  |
>
> The results, presented in the table, reveal several clear trends. The outcome-only reward drives the orchestrator to achieve higher task performance but at greater cost, whereas the outcome + efficiency configuration teaches the model to strike a more effective balance between performance and resource usage. When the preference for API-based tools is introduced, the proportion of API-based tool calls increases substantially; similarly, the model shifts toward Qwen-series models when trained with the corresponding preference reward. Notably, the comprehensive multi-objective setting demonstrates a clear advantage over all other reward configurations measured by preference-aware rewards. Taken together, these results underscore the benefit of multi-objective optimization, which enables the model to jointly optimize for performance, efficiency, and user preferences — an outcome that simpler, single-objective reward designs cannot achieve.
>
> We will incorporate these discussions in the revised manuscript. Thank you for your valuable feedback, which significantly enhances our paper's clarity and contribution.

---

> > ### Author Rebuttal · Reviewer_KSvJ · 2026-04-03
> >
> > Satisfied with the additional context and results

---

> > > ### Author Response · Authors · 2026-04-06
> > >
> > > It’s wonderful to hear that your concerns have been adequately addressed and you are satisfied with the additional context and results! Thank you so much for your valuable comments to our paper!

---

### Official Review · Reviewer_DTbX · 2026-03-12

**Soundness:** 3
**Presentation:** 3
**Significance:** 3
**Originality:** 3
**Overall Recommendation:** 5
**Confidence:** 4

**Summary:**

This paper introduces ToolOrchestra, a pipeline for training a small LLM to act as an orchestrator that efficiently coordinates a diverse set of basic tools, specialized models, and powerful generalist LLMs. The authors introduce ToolScale, an automated data synthesis pipeline to generate high-quality, multi-turn tool-use training data. Besides, GRPO is utilized to train the model with the designed reward systems, including outcome-based reward, efficiency, and preference. Extensive experiments demonstrate that the Orchestrator-8B with massive tools and models outperforms base models like GPT-5 and Claude Opus 4.1 on complex reasoning benchmarks (HLE, FRAMES, and $\tau^2$-Bench) while utilizing only a fraction of the cost and wall-clock time.

**Compliance With Llm Reviewing Policy:**

Affirmed.

**Key Questions For Authors:**

Please refer to the weaknesses.

**Limitations:**

yes

**Strengths And Weaknesses:**

**Strengths:** The paradigm of using a small model to orchestrate larger, more intelligent models is highly inspiring. It aggregates the advantages of different tools and models, showing promising performance with lower cost.

**Weaknesses*:*

1. While the paper introduces the ToolScale dataset and provides a complex 12-step example in the Appendix, the overall statistical distribution of task difficulty remains unclear. The authors should provide detailed metrics on the constructed problems, such as the average number of reasoning steps and the average number of distinct tools required to solve a task.
2. Generalization to Newly Added Tools: Specifically, if a new model or tool is added to the pool at test time, will the orchestrator underutilize it because it hasn't learned its value during the RL exploration phase? The paper should discuss whether the tools in training must strictly align with testing, and how the model builds trust in newly plugged-in tools zero-shot. Besides, the performance in Table 2 is largely behind that in Table 1. I wonder whether it is because the super model (i.e., GPT-5) is not in the tool configuration.
3. The current evaluation on HLE, FRAMES, and $\tau^2$-Bench is strong but primarily focuses on QA, RAG, and conversational API calling. To further demonstrate the effectiveness of the Orchestrator, the authors should consider evaluating on more diverse and interactive agentic benchmarks. For instance, BrowseComp (need more turns of interaction).

---

> ### Author Rebuttal · Authors · 2026-03-31
>
> We appreciate the reviewer's thoughtful and valuable feedback. We are pleased that the reviewer considers our approach of using a smaller language model to coordinate larger, more capable models to be highly compelling, and that it effectively combines the strengths of various tools and models while delivering strong performance at reduced cost. Below, we address individual questions raised in the review.
>
> **ToolScale statistics**
> We appreciate the reviewer's inquiry regarding the detailed metrics of the synthesized datasets. We present the 25th, 50th, and 75th percentile values for both the number of reasoning steps and the number of distinct tools employed by GPT-5 to complete a task. These statistics demonstrate that the ToolScale dataset maintains a well-balanced mix of tasks across varying difficulty levels.
>
> |                               | Min | 25th | 50th | 75th | Max |
> | ----------------------------- | --- | ---- | ---- | ---- | --- |
> | The number of reasoning steps | 3   | 6    | 10   | 13   | 20  |
> | The number of distinct tools  | 2   | 4    | 6    | 9    | 14  |
>
>
> **Generalization to newly added tools**
> As we mentioned in section 3.1, tools are specified in JSON as a list of objects, and we curate a diverse set of tool configurations during training (section 3.3). This means new tools or models can be easily integrated into the orchestration framework simply by supplying their configurations. The LLM descriptions are based on 10 task trajectories that capture both strengths and weaknesses. Through reinforcement learning, the orchestrator learns to follow these configurations when selecting tools. Because training involves a broad range of tools, this ability generalizes well — allowing the model to effectively utilize unfamiliar tools at test time based solely on their configuration details. The weaker results in Table 2 stem not only from GPT-5's absence, but from the overall lower quality of the available tool set, including less powerful models like Codestral-22B-v0.1 and OpenMath-Llama-2-70b. Despite this, Orchestrator-8B still delivers notable gains in both accuracy and efficiency under these conditions, demonstrating its strong generalization and the value of the orchestration approach.
>
>
> **Results in BrowserComp**
> As recommended by the reviewer, we evaluated Orchestrator-8B alongside GPT-5 on the BrowseComp benchmark. The findings demonstrate that our Orchestrator-8B continues to perform strongly on this interactive, agent-based evaluation.
>
> |                 | Accuracy |
> | --------------- | -------- |
> | GPT-5           | 54.9     |
> | Orchestrator-8B | 58.2     |
>
>
> We will include all of the above discussions in our revised manuscript. Thank you again for your valuable feedback, which will significantly enhance our manuscript’s clarity and contribution.

---

> > ### Author Rebuttal · Reviewer_DTbX · 2026-04-03
> >
> > Thanks authors for their responses. My concerns have been adequately addressed.

---

> > > ### Author Response · Authors · 2026-04-06
> > >
> > > We would like to thank you again for your efforts and positive feedback!
> > >
> > > We are very happy that our response and updated presentation have adequately resolved your concerns. Your thoughtful comments have contributed significantly to refining our presentation and improving the overall clarity of the manuscript.
> > >
> > > Thank you!

---

### Official Review · Reviewer_WKgT · 2026-03-13

**Soundness:** 2
**Presentation:** 3
**Significance:** 3
**Originality:** 2
**Overall Recommendation:** 4
**Confidence:** 3

**Summary:**

This paper introduces ToolOrchestra, a reinforcement learning framework for training a small (8B parameter) language model to serve as an orchestrator that coordinates a heterogeneous pool of tools, including basic utilities (web search, code interpreters, domain-specific APIs), specialized expert LLMs (e.g., math or coding models), and generalist frontier LLMs (e.g., GPT-5, Claude Opus 4.1). The orchestrator is trained end-to-end via Group Relative Policy Optimization (GRPO) with a composite reward signal that jointly accounts for task outcome correctness, computational efficiency (cost and latency), and alignment with user-specified tool preferences. To support RL training, the authors also contribute ToolScale, a synthetic data generation pipeline that produces verifiable multi-turn tool-use tasks across 10 domains. The resulting Orchestrator-8B is evaluated on three challenging benchmarks: Humanity's Last Exam (HLE), FRAMES, and tau2-Bench. The headline claim is that Orchestrator-8B achieves 37.1% on HLE versus GPT-5's 35.1% with basic tools, while being roughly 2.5 times more cost-efficient, and similarly outperforms baselines on the other two benchmarks. The paper also reports generalization to unseen tool pools and pricing regimes, controllability via user preference instructions, and adversarial robustness.

**Compliance With Llm Reviewing Policy:**

Affirmed.

**Final Justification:**

The rebuttal addressed all my concerns, specifically by including experiments with prior art and performing a reward ablation. I am less confident about my final review now, however. I still think there are weaknesses relating to novelty and that is why I didn't choose 'Accept'.

**Key Questions For Authors:**

1. What fraction of Orchestrator-8B's correct answers on HLE come directly from a single GPT-5 tool call with minimal additional reasoning? This would clarify whether the orchestrator is performing meaningful multi-step orchestration or primarily acting as a router to GPT-5. If a large fraction of correct answers are simple pass-throughs, the claimed orchestration benefits are substantially weaker.

2. Have you tried training and evaluating with a judge model that is fully independent from the tool set (e.g., Claude or Gemini as judge, with GPT-5 only as a tool)? A positive result with an independent judge would substantially mitigate the circularity concern and would likely increase my confidence in the evaluation. A negative result (substantially different performance) would suggest that the current numbers are inflated by judge-tool alignment.

3. Can you provide ablations of the reward components (outcome-only, outcome + efficiency, outcome + preference, full reward)? This is necessary to understand whether the multi-objective reward design is actually driving improvements. If outcome-only reward achieves similar performance, the claimed contribution of the reward design is diminished; if the full reward meaningfully outperforms, the contribution is validated.

4. How does Router-R1 (Zhang et al., NeurIPS 2025) compare to your method, both methodologically and empirically? The approaches appear to share significant overlap in formulation (RL for multi-LLM routing, cost-aware rewards, generalisation to unseen models). If Orchestrator substantially outperforms Router-R1, that strengthens the paper; if not, the novelty claim needs revision.

5. What happens when GPT-5 is completely removed from the tool set at evaluation time? Measuring the resulting accuracy drop would quantify how much the system depends on its strongest single tool versus benefiting from genuine orchestration across the full pool.

**Limitations:**

The authors discuss limitations briefly in the impact statement (Section 9), noting risks of improper tool selection and the need for responsible deployment. However, the discussion is quite general and does not address several specific limitations that the paper itself reveals: the 35% failure rate on adversarial prompts, the dependence on proprietary APIs for both training and evaluation, the unquantified training cost, and the bounded nature of orchestrated performance relative to the strongest available tool. Moving some of the limitation discussion from scattered appendix sections into the main text would improve transparency.

**Strengths And Weaknesses:**

## Strengths and Weaknesses

### Strengths

**S1. Important and timely problem formulation.** The paper addresses a central question for agentic LLM systems: how should a system choose among tools and models of varying cost and capability? This orchestration approach is more realistic than the common setting where a single model has access to a fixed set of basic tools. The practical thesis that a small, cheap model can serve as an intelligent router over expensive and specialised tools is compelling and directly relevant to deployment scenarios where these savings are important.

**S2. Well-designed multi-objective reward.** The three-component reward function integrating outcome correctness, efficiency penalties (monetary cost and latency), and user preference alignment is a sensible and well-motivated design. Most prior tool-use RL work focuses narrowly on accuracy; incorporating cost and controllability makes this work substantially more practical. The preference vector mechanism allowing users to specify tradeoffs at inference time (Table 18) is a genuinely useful feature that goes beyond what most comparable systems offer.

**S3. Compelling empirical results.** The headline numbers are strong: 37.1% on HLE, 76.3% on FRAMES, and 80.2% on tau2-Bench, all at substantially lower cost and latency than baselines using the expanded tool set. The cost-performance Pareto curves (Figure 5) clearly demonstrate that Orchestrator dominates baselines across budget levels. The breadth of evaluation is commendable and includes generalisation to unseen tool pools (Table 2), unseen pricing regimes (Table 3), preference-following evaluation (Table 18 and Table 21), filtering ablations (Figure 6), scaling with orchestrator size (Table 19), reasoning effort analysis (Table 22), and adversarial robustness (Table 16).

**S4. Valuable diagnostic insight on prompted orchestrators.** The observation that off-the-shelf LLMs exhibit systematic biases when used as orchestrators via prompting alone is a useful finding. Specifically, that GPT-5 disproportionately delegates to GPT-5-mini (self-enhancement bias), while Qwen3-8B defaults heavily to GPT-5 (other-enhancement bias). This observation provides strong motivation for dedicated RL training and is one of the paper's most convincing contributions.

**S5. Useful data contribution.** The ToolScale synthesis pipeline and the resulting dataset across 10 domains address a real gap in verifiable multi-turn tool-use training data. The three-criterion evaluation scheme (execution correctness, process fidelity, operation completeness) for trajectory verification is well-designed and helps ensure high-quality reward signals during RL training. The commitment to release this dataset publicly is valuable for the community.

**S6. Training stability contributions.** The three filtering strategies (homogeneity, format consistency, invalid output filtering) described in Section 3.2 are practical contributions to stabilising GRPO training in multi-turn agentic settings. Figure 6 convincingly shows that all three are needed to sustain reward improvement.

### Weaknesses

**W1. Circularity in evaluation: GPT-5 as both tool and judge.** This is the most significant methodological concern. GPT-5 is simultaneously used as (a) a tool available to the orchestrator, (b) the reward judge during RL training, and (c) the correctness evaluator at test time. This risks self bias: the orchestrator is optimised to produce outputs that GPT-5 judges as correct, and GPT-5 is also one of the tools it can call. If the orchestrator learns to route to GPT-5 and pass through its answers, the same model is both generating and judging the output, which could artificially inflate performance metrics. While Table 17 shows that Qwen3-32B has only a slightly higher error rate as a judge (2-4%), the training itself used GPT-5 as judge, so the potential bias is baked into the learned policy. The paper would be substantially strengthened by reporting results with a fully independent judge (e.g., Claude or Gemini) and by reporting the fraction of correct answers that originated from a single GPT-5 tool call.

**W2. Baseline comparisons are not fully fair.** Several aspects of the experimental design disadvantage the baselines. Most importantly, the comparison pits a trained orchestrator against zero-shot prompted baselines. The authors do not attempt to improve the prompted orchestrators via better prompting strategies (e.g., few-shot demonstrations of balanced tool use, explicit cost constraints in the system prompt). Additionally, GPT-5 as a prompted orchestrator with expanded tools performs substantially worse than GPT-5 with only basic tools (21.2 vs. 35.1 on HLE), which is attributed to inherent biases but could potentially be mitigated with better prompt engineering. RestGPT uses default configuration rather than being tuned for the expanded tool setting. RouteLLM is designed for binary routing between two models, making its inclusion as a baseline somewhat misleading in a multi-tool setting. A critical missing baseline is Router-R1 (Zhang et al., NeurIPS 2025), which formulates multi-LLM routing and aggregation as a sequential decision process with RL, employs cost-aware rewards, and conditions on model descriptors for generalisation to unseen models. This is highly relevant and should have been discussed and compared against!

**W3. Missing reward component ablations.** The paper proposes a three-part reward (outcome + efficiency + preference) as a central methodological contribution, but never cleanly ablates these components. The filtering ablation (Figure 6) addresses training stability, not the reward design. Without experiments showing results for outcome-only reward, outcome + efficiency, outcome + preference, and variations in the preference weighting, it is impossible to determine how much the multi-objective reward is actually driving the improvements versus a simpler reward design. This is a significant gap given that the reward formulation is presented as one of the paper's main contributions.

**W4. Reproducibility concerns due to proprietary dependencies.** A large fraction of the system's value comes from orchestrating proprietary tools and models such as GPT-5 and Claude Opus 4.1, and GPT-5 also serves as the default judge. This creates multiple reproducibility issues: API pricing changes could affect reported results, model behaviour may drift over time due to provider updates, and other researchers cannot fully replicate the setup. The paper would benefit from including more open-source-only experiments that demonstrate the method's value independently of proprietary components.

**W5. HLE comparison requires clarification.** The footnote in Table 1 notes that the "Existing reported SOTA" HLE results are based on the full set, while other baselines and Orchestrator-8B are evaluated only on the text-only subset. This makes the 37.1% vs. 35.1% comparison potentially misleading if readers do not notice the footnote. The paper should state this caveat prominently and ensure that all compared numbers are on exactly the same evaluation set.

**W6. Incomplete disclosure of training cost.** While inference cost is thoroughly analysed, the training cost of Orchestrator-8B is not fully specified. The paper mentions 16 H100 GPUs but does not state training duration, total GPU-hours, the compute cost of generating ToolScale data, or the number of RL training steps beyond what can be inferred from Figure 6. A model that costs substantially more to train but saves at inference may not be worthwhile for all use cases, and full cost accounting is needed for the reader to judge this tradeoff.

**W7. Limited analysis of failure modes and orchestration depth.** The paper lacks analysis of when and how orchestration fails. It does not report what fraction of correct answers come from a single GPT-5 tool call (i.e., simple routing rather than meaningful multi-step orchestration), nor does it analyse the distribution of tool calls per successful versus failed trajectory, or systematic task types where the 8B model makes poor routing decisions. Relatedly, the framing of "elevating intelligence" in the title overstates the contribution: Orchestrator's performance is fundamentally bounded by the capabilities of its tool models, and the 2-point improvement over GPT-5 with basic tools on HLE is better characterised as efficient composition than intelligence elevation.

**W8. Moderate methodological novelty.** While the paper is strong as a systems and empirical contribution, the algorithmic novelty is moderate. Framing tool use as sequential decision-making with RL is established; using verifiable reward signals and efficiency penalties is now common in recent agent RL work (Search-R1, OTC, ToRL); routing between strong and weak models has prior art; and synthetic environment construction for multi-turn RL agents is becoming standard. The contribution is best understood as a comprehensive integration and scaling-up of several known ideas into a heterogeneous orchestration setting, rather than a fundamentally new learning principle. The paper should position itself more carefully in this regard.

### Minor Issues

- The "self-enhancement bias" terminology references Zheng et al. (2023), which discusses LLM-as-judge biases, not the orchestrator preference bias described here. The authors should either coin their own term with proper attribution or find a more appropriate reference, such as "self-attribution bias" (Khullar et al. 2026), which shows models are biased when scoring their own actions.
- Notation is inconsistent: the MDP formulation defines user preferences as p = (0 <= p_a <= 1) but later these become a vector P with a different structure.
- All reported results appear to be single runs (temperature=0). While deterministic inference reduces variance, RL training is stochastic, and reporting results across multiple seeds would strengthen the claims.
- The adversarial robustness results (65.4% rejection rate) mean roughly 35% of adversarial prompts succeed, which raises safety concerns for an orchestrator that can invoke powerful models and execute code. The impact statement acknowledges risks but does not propose concrete mitigation strategies.
- Several figures (especially the pie charts in Figure 4) are difficult to read due to insufficiently distinct colour schemes.
- Latency measurement in wall-clock minutes depends on API response times, network conditions, and server load, making this metric inherently noisy and hard to reproduce.
- The GeneralThoughtArchive dataset used in training is referenced but not described in sufficient detail regarding its contents, size, or interaction with ToolScale during training.

---

> ### Author Rebuttal · Authors · 2026-03-31
>
> We thank the reviewer for the thorough and constructive feedback on our paper. Below we address the questions raised in the review. Due to the space limit, the additional experimental results and details are omitted and could be provided in the latter discussion period.
>
> **GPT-5 as both tool and judge**
> By replacing GPT-5 with Claude 4.6 Opus as an independent judge, we find that the result difference is within 0.1
>
> **Baseline comparison**
> To ensure fair comparison, we use consistent prompts across all evaluated models. Adding few-shot demonstrations and explicit cost constraints into the prompts. brings no significant improvement in performance (less than 1%) and efficiency (less than 2 points), which suggest that prompt engineering alone may not be sufficient to explain the poor results of GPT-5 in this setting.
>
> Regarding RestGPT, as discussed in Section 4.3, we evaluate it in both the default setting (basic tools only) and an expanded setting (basic tools plus specialized/generalist LLMs). For RouteLLM, we agree that its inclusion may cause confusion given its design for binary routing, and we will consider removing it from Table 1 in a revised version.
> Following the reviewer's suggestion, we have also added Router-R1 as an additional baseline, which achieve 31.8 on HLE, 71.5 on FRAMES and 72.2 on  $\tau^2$-Bench. We note, however, that this comparison is not entirely apples-to-apples: the publicly released Router-R1 model has only 3 billion parameters and was not trained with tools such as web search or code interpreters. That said, the results show that Router-R1 is a strong baseline that significantly outperforms GPT-5 when using LLMs as tools.
>
> **Error analysis**
> Through detailed analysis of Orchestrator-8B's output trajectories, we identified two typical failure modes: (1). Retrieval trap: wrongly route queries to web search when a reasoning-first approach would be more appropriate; (2). Cascading Error Amplification in Multi-Turn Tool Chains: an error introduced at any stage can propagate to, or even amplified by downstream steps. We will add more details and examples in our camera-ready version.
>
> **Novelty**
> We agree with the reviewer that prior work has made significant strides in areas such as sequential decision-making, verifiable rewards with efficiency penalties, etc. However, we want to highlight a key contribution of our work: we show that a small language model can serve as an orchestrator for a diverse set of tools — including ones that are far more capable and intelligent than itself — while achieving state-of-the-art results (also noticed by the reviewer DTbX). To our knowledge, this is the first demonstration of this kind, and we believe it carries meaningful long-term implications, particularly for scenarios where a less capable entity needs to effectively leverage much more powerful tools or agents (for instance, humans manipulating highly knowledgeable LLMs). We see this paper as an initial proof of concept showing that small language models can develop such orchestration abilities through RL training, with future research offering deeper, more systematic exploration of this paradigm. Beyond this core contribution, we are also excited about several additional points: (1) our proposed orchestration framework unifies tools and models in the same space, and the RL-trained Orchestrator-8B reduces the cost of solving complex agentic tasks by 2.5×, (2). the synthesized data and generation pipeline serve as a useful resource to the community, and (3) the Orchestrator-8B exhibits a strong ability to adapt to user preferences across varied tool- and model-use scenarios — a property that is critical for real-world deployment across industry and the broader community.
>
>
> **Others**
> Due to the space limit, we briefly discuss other points here: (1). We train Orchestrator-8B for 40 hours, amounting to 640 GPU hours in total, and the ToolScale is generated at a cost of roughly $400 and 50 GPU hours; (2). The variance of multiple run with different seeds is within 1%; (3). We generate additional safety data using seed  instructions from [1], and could increase the robustness score (described in Appendix N) to over 90 without sacrificing benchmark performance; (4). only 4.6% in HLE and 3.3% FRAMES of correct answers originate from a single GPT-5 call, and the noticeable difference in tool call distribution of failed trajectory is the increased usage of GPT-5, where it makes additional attempt to solve difficult problems; (5). removing GPT-5 from tool candidates results in up to 5% performance degradation; (5). We will address the minor issues mentioned by the reviewer.
>
> [1]. https://github.com/ChenWu98/agent-attack
>
> The reviewer could find more discussions on reward ablation and HLE comparison in our reply to the reviewer KSvJ, and dependency on proprietary models in our reply to the reviewer mamh.

---

> > ### Author Rebuttal · Reviewer_WKgT · 2026-04-05
> >
> > Thank you for your clarification - I like the paper and will increase my score accordingly.

---

> > > ### Author Response · Authors · 2026-04-06
> > >
> > > It's great to know that your concerns have been addressed to your satisfaction and that you're updating your rating as a result! Thank you!

---

### Official Review · Reviewer_mamh · 2026-03-13

**Soundness:** 3
**Presentation:** 3
**Significance:** 3
**Originality:** 3
**Overall Recommendation:** 5
**Confidence:** 3

**Summary:**

This paper presents ToolOrchestra, a framework that trains a small language model as an orchestrator to coordinate heterogeneous tools and LLMs for solving complex multi-step agentic tasks. In this framework, LLMs are treated as tools and accessed by the orchestrator through predefined interfaces. The orchestrator is trained using RL with three types of rewards: task outcome correctness, computational cost and efficiency, and alignment with user preferences. Experiments show that the resulting Orchestrator-8B achieves higher performance than several baselines, including GPT-5, on HLE, FRAMES, and τ²-Bench, while requiring lower inference cost.

**Compliance With Llm Reviewing Policy:**

Affirmed.

**Final Justification:**

I thank the authors for their detailed response, which has addressed my concerns, and I will maintain my positive score.

**Key Questions For Authors:**

How are the weights for different reward signals (outcome, efficiency, and user preference) determined? Was any systematic ablation study conducted on these weights?

**Limitations:**

yes

**Strengths And Weaknesses:**

Strengths:

1. Treating LLMs as tools that can be orchestrated alongside other tools is a neat idea with more realistic deployment scenarios compared to prior work.

2. Beyond the typical focus on reasoning accuracy, the reward design also accounts for time efficiency, cost, and user preference, which makes the framework more practically grounded.

3. Some useful tricks are introduced, such as using LLM-generated summaries for tool descriptions and randomly sampling tool subsets per training example. The paper also includes experiments validating generalization, which is good to see.


Weaknesses:

1. The paper heavily relies on GPT-5 for training data synthesis and reward judgment. This significantly limits reproducibility.

2. A large portion of the training data is synthetically generated by LLMs. This may limit the model’s robustness in noisy real-world environments, which appears somewhat inconsistent with the paper’s motivation of improving the practicality of agentic systems by considering efficiency, cost, and user preference.

3. The paper mentions that existing SOTA results on HLE are reported on the full set, while Orchestrator-8B is only evaluated on the text-only subset. This inconsistency makes the comparison less convincing (37.1% vs. 35.1%). It would be better to re-run or re-report results under a unified evaluation setup.

4. There are a few typos in the paper, e.g., "enfficiency" and "compex" in the conclusion section.

---

> ### Author Rebuttal · Authors · 2026-03-31
>
> Thank you for your review and important comments to the paper. We are happy to hear that the reviewer finds the idea to treat LLMs as tools neat, the framework with novel reward design practically grounded, and the training settings with generalization validation effective. Below we address the concerns and questions raised in the review.
>
> **Dependency on GPT-5 and synthetic data**
> We appreciate the reviewer's concern regarding reproducibility and would like to address it from several angles.
> First, our use of synthetic data and strong proprietary models during development follows established practice in the research community [1][2], where such models are commonly employed for data synthesis and evaluation. The specific model used to execute the pipeline is an implementation detail that can be readily substituted, and the synthetic data we release serves primarily as a practical resource for the open-source community.
> Second, it is worth clarifying that ToolScale constitutes only a portion of our training data. We also incorporate GeneralThought [3], a realistic dataset covering diverse and comprehensive user scenarios, which further reduces our dependence on synthetic data generated by GPT-5.
> That said, we recognize that the reviewer raises a valid point: demonstrating that our pipeline works without reliance on proprietary models would meaningfully strengthen both its reproducibility and practical value. To this end, we have conducted additional experiments in which we replace GPT-5 with open-source alternatives across the entire pipeline. Specifically, we use Kimi-K2.5 for data synthesis, Qwen-3.5 (397B, A17B) and gpt-oss-20b as specialized and generalist LLMs (replacing GPT-5 and GPT-5-mini in the paper), and Qwen-3.5 (35B, A3B) for reward judgment. We also retain GeneralThought and incorporate realistic tool-use training data from [4], both of which help ground the training process in realistic user scenarios. In the table below, the first two rows are consistent with the paper, while the last two rows reflect the fully open-source configuration using basic tools, specialized LLMs, and generalist LLMs.
>
> |                               | HLE  | FRAMES | $\tau^2$-Bench | cost | latency |
> | ----------------------------- | ---- | ------ | --------------- | ---- | ------- |
> | GPT-5                         | 21.2 | 57.5   | 62.3            | 17.8 | 13.6    |
> | Orchestrator-8B (GPT-5)       | 37.1 | 76.3   | 80.2            | 9.2  | 8.2     |
> | Qwen-3.5 (397B, A17B)         | 41.6 | 72.6   | 81.4            | 5.6  | 4.8     |
> | Orchestrator-8B (open-source) | 52.4 | 80.2   | 89.5            | 2.5  | 2.2     |
>
> As shown, our Orchestrator-8B (open-source) demonstrates significant improvements over Qwen-3.5 given the same set of tools, confirming the general applicability of our approach beyond proprietary models.
> We will update the manuscript to include these results and highlight the open-source pipeline as a recommended configuration for reproducibility.
>
> [1]. Self-Instruct: Aligning Language Models with Self-Generated Instructions, https://arxiv.org/abs/2212.10560 \
> [2]. WizardLM: Empowering large pre-trained language models to follow complex instructions, https://arxiv.org/abs/2304.12244 \
> [3]. https://huggingface.co/datasets/natolambert/GeneralThought-430K-filtered \
> [4]. https://github.com/OpenBMB/ToolBench
>
>
> **HLE comparison**
> Please see our reply to the reviewer KSvJ on HLE comparison.
>
> **Typos**
> We thank the reviewer for carefully checking our manuscript. We will update it in the camera-ready version.
>
> **Ablation study on reward components**
> Please see our response to the reviewer KSvJ on Ablation study on reward components.

---

> > ### Author Rebuttal · Reviewer_mamh · 2026-04-03
> >
> > I thank the authors for their detailed response, which has addressed my concerns, and I will maintain my positive score.

---

> > > ### Author Response · Authors · 2026-04-06
> > >
> > > We are glad to hear that our detailed response has addressed your concerns. Thank you so much for your positive feedback and your time and efforts in reviewing our paper!

---

### Decision · Program_Chairs · 2026-04-30

**Decision:**

Accept (regular)

**Comment:**

This paper studies training a small language model to orchestrate heterogeneous tools and stronger models for efficient agentic problem solving. Reviewers agreed that it addresses an important problem and provides a strong empirical contribution, with convincing results on performance-cost tradeoffs and broad evaluation.

The main concerns involved evaluation methodology, reproducibility, and novelty/positioning. After rebuttal, I find that these concerns were substantially addressed through additional experiments and clarifications, including independent judging, reward ablations, stronger baseline comparisons, and clarification of the HLE setup. Reviewer discussion after rebuttal was clearly positive.

While the methodological novelty is somewhat more moderate than the strongest framing may suggest, the paper is technically solid, timely, and practically relevant. I therefore recommend acceptance.